# The Complexity of the Super Subdivision of Cycle-Related Graphs Using Block Matrices

**Mohamed R. Zeen El Deen *** [ID], **Walaa A. Aboamer** [ID] **and Hamed M. El-Sherbiny** [ID]

Department of Mathematics and Computer Science, Faculty of Science, Suez University, Suez 43111, Egypt;
walaa.abdelrazek@sci.suezuni.edu.eg (W.A.A.); h_elsherbiny@hotmail.com (H.M.E.-S.)
* Correspondence: mohamed.zeeneldeen@sci.suezuni.edu.eg or mohamed.zeeneldeen@suezuniv.edu.eg;
  Tel.: +20-100-903-9449

**Abstract:** The complexity (number of spanning trees) in a finite graph $\Gamma$ (network) is crucial. The quantity of spanning trees is a fundamental indicator for assessing the dependability of a network. The best and most dependable network is the one with the most spanning trees. In graph theory, one constantly strives to create novel structures from existing ones. The super subdivision operation produces more complicated networks, and the matrices of these networks can be divided into block matrices. Using methods from linear algebra and the characteristics of block matrices, we derive explicit formulas for determining the complexity of the super subdivision of a certain family of graphs, including the cycle $C_n$, where $n = 3, 4, 5, 6$; the dumbbell graph $Db_{m,n}$; the dragon graph $P_m(C_n)$; the prism graph $\Pi_n$, where $n = 3, 4$; the cycle $C_n$ with a $P_{\frac{n}{2}}$-chord, where $n = 4, 6$; and the complete graph $K_4$. Additionally, 3D plots that were created using our results serve as illustrations.

**Keywords:** complexity; super subdivision; dragon graph; cycle with a chord; dumbbell graph

## 1. Introduction

Graph theory is a theory that merges mathematics with computer technology. For a simple, undirected and connected graph $\Gamma = (V(\Gamma), E(\Gamma))$, *a spanning tree* is a subset of the edges of $\Gamma$ that connect all the vertices together without any cycles [1]. *The complexity* of $\Gamma$, denoted by $\tau(\Gamma)$, is the total number of spanning trees existent therein. It is used to calculate the connected and acyclic components that are present in it. Spanning trees are well recognised, and several studies have been conducted to prove their existence and count all of their numbers. One can introduce the complexity function $\tau(\Gamma) = \tau(\kappa)$ for an infinite family of graphs $\Gamma_\kappa$, $\kappa \in \mathbb{N}$, which makes it much easier to calculate and identify the number of corresponding spanning trees, especially when these numbers are very large. The number of spanning trees is employed in a variety of fields, including engineering and network reliability. This invariantly contributes to enhancing the robustness of wireless sensor networks (WSNs) and other analogous mobile networks. The security plan for a building's sensitive area is an example of another way complexity is used.

One of the biggest chemistry challenges is correctly recognising a chemical molecule. In [2], Joita et al. introduced graph representations of molecules as well as answers to questions about relationships between the structure of chemical compounds and various parameters. The number of spanning trees is employed in the field of chemistry. In [3,4], Nikolić et al. studied the complexity of molecules using approaches based on the topological complexity, that is, the complexity of the corresponding molecular graphs. Several measures of the topological complexity, such as those introduced by Bertz and Randić or based on the number of spanning trees, have been studied and a comparison was made between these measures of topological complexity for selected molecular graphs. For more details, please refer to [5–7].

A different way of displaying and condensing data from graphs is via matrices. The same information is contained in both a matrix and a graph, but a matrix is more effective for computing and computer analysis [8]. Given a graph $\Gamma = (V(\Gamma), E(\Gamma))$ with $|V(\Gamma)| = \kappa$, the *adjacency matrix* of $\Gamma$ denoted by $A(\Gamma) = a_{\ell m}$ is a $\kappa \times \kappa$ matrix defined as follows. The rows and the columns of $A(\Gamma)$ are indexed by $V(\Gamma)$. If $\ell \neq m$, then the $(\ell, m)$ -entry of $A(\Gamma)$ is 0 for nonadjacent vertices $\ell$ and $m$, and the $(\ell, m)$-entry is 1 for adjacent $\ell$ and $m$. The $(\ell, \ell)$-entry of $A(\Gamma)$ is 0 for $\ell = 1, \cdots, \kappa$, i.e.,

$$a_{\ell m} := \begin{cases} 1 & \text{if} \quad v_\ell v_m \in E(\Gamma) \,, \\ 0 & \text{if} \quad \text{otherwise.} \end{cases}$$

The *degree matrix* [8] $D$ for $\Gamma$ is a $\kappa \times \kappa$ diagonal matrix defined as

$$D_{\ell m} := \begin{cases} deg(v_\ell) & \text{if} \quad \ell = m \,, \\ 0 & \text{if} \quad \text{otherwise.} \end{cases}$$

where the vertex's degree $deg(v_\ell)$ is the number of times an edge ends at that vertex.

The Laplace matrix of $\Gamma$ denoted by $L(\Gamma)$ is defined as $D(\Gamma) - A(\Gamma)$. It is well known that $L(\Gamma)$ is a positive semidefinite matrix with the smallest eigenvalue 0. Kirchhoff presented the first method to calculate the number of spanning trees defined as the matrix tree theorem [9], which says that all the cofactors of $L$ are equal and their common value is equal to the complexity $\tau(\Gamma)$ of $\Gamma$.

Temperley [10] has shown that

$$\tau(\Gamma) = \frac{1}{\kappa^2} \, det( \, L \, + \, J \,), \tag{1}$$

where $J$ is a $\kappa \times \kappa$ matrix, all of whose elements are ones.

Brownaj et al. [11] applied various methods to calculate the complexities of graphs that represent fullerenes ($C_{60}$, $C_{70}$ and $C_{120}$ molecules). These graphs are large, regular and highly symmetrical. Kirby et al. [12] presented a theorem called the cycle theorem, by means of which the complexity of a labelled planar or non-planar graph may be calculated from, in general, two determinants. These are much smaller than in the traditional matrix tree theorem. They applied this theorem to a conventional toroidal polyhex.

The contraction–deletion theorem is one of the most often used techniques for determining complexity. The complexity $\tau(\Gamma)$ of a graph $\Gamma$ is equal to $\tau(\Gamma) = \tau(\Gamma - x) + \tau(\Gamma/x)$, where $x$ is any edge of $\Gamma$, $\Gamma - x$ is the deletion of $x$ from $\Gamma$ and $\Gamma/x$ is the contraction of $x$ in $\Gamma$. This provides a recursive way to determine how complex a graph is, see [13].

Another significant technique is using electrically equivalent transformations of networks. Zhang et al. [14] discovered a clear-form formula for the enumeration of spanning trees of the subdivided line graph of a simple connected graph using the theory of electrical networks. Teufl and Wagner [15] have demonstrated that the number of spanning trees in a network only varies by a factor if any of its subnetworks are replaced by an electrical equivalent network. This is crucial information that makes it simple to calculate the number of spanning trees in a network [16].

Zeen El Deen et al. [17] created a clear computation of the complexity of a duplicate graph using the splitting, shadow, mirror and total operations of a certain family of graphs. The authors presented nine network designs in [18], built using squares of various average degrees, and established a more precise, simple formula for the number of spanning trees in each of these networks. In [19], Zeen El Deen examined the complexity of several classes of prisms of graphs with connections to paths and cycles. For several families of graphs, counting and maximising the number of spanning trees has recently been the subject of numerous studies [20–25].

The complement of a graph $\Gamma$ is a graph $\overline{\Gamma}$ on the same vertices such that two distinct vertices of $\overline{\Gamma}$ are adjacent if and only if they are not adjacent in $\Gamma$. That is, to create the

complement of a graph, one removes all edges that were previously there and adds all edges that are needed to build a complete graph.

**Lemma 1** ([26]). *Let $\Gamma$ be a graph with $\kappa$ vertices. Then,*

$$\tau(\Gamma) = \frac{1}{\kappa^2} \, det \, (\, \kappa \, I - \overline{D} + \overline{A} \,), \tag{2}$$

*where $\overline{D}$, $\overline{A}$ are the degree and adjacency matrices, respectively, of $\overline{\Gamma}$, where $\overline{\Gamma}$ is the complement of $\Gamma$.*

A matrix that has been broken up into blocks that are themselves matrices is known as a block matrix. The matrix is divided by making one or more vertical or horizontal cuts across it [27]. Block matrices are crucial for determining the number of spanning trees in graphs.

**Lemma 2** ([28,29]). *Suppose $\Phi$, $\Delta$, $\Omega$ and $\Psi$ are block matrices of dimension $\ell \times \ell$ and $\Phi$ is invertible. Then,*

$$(i) \quad det \begin{pmatrix} \Phi & \Delta \\ \Omega & \Psi \end{pmatrix} = det \, (\Phi) \, \times \, det \, (\Psi - \Omega \, \Phi^{-1} \Delta).$$

$$(ii) \quad det \begin{pmatrix} \Phi & \Psi \\ \Psi & \Phi \end{pmatrix} = det \, (\Phi + \Psi) \, \times \, det \, (\Phi - \Psi).$$

**Lemma 3** ([8]). *Let $E_\ell(\xi)$ be the $\ell \times \ell$ circulant matrix given by*

$$E_\ell(\xi) = \begin{pmatrix} \xi & 1 & 1 & 1 & \dots & 1 \\ 1 & \xi & 1 & 1 & \dots & 1 \\ \vdots & \vdots & \dots & \ddots & \ddots & \vdots \\ 1 & 1 & 1 & \dots & 1 & \xi \end{pmatrix}_{\ell \times \ell} , \textit{ then } det \, [E_\ell(\xi)] = (\xi + \ell - 1) \, (\xi - 1)^{\ell - 1}.$$

Graph operations [30] create new graphs from old ones; the super subdivision operation is employed currently.

**Definition 1.** *In a complete bipartite graph $K_{2,t}$ , the part consisting of two vertices is referred as the two-vertex part of $K_{2,t}$ , and the part consisting of $t$ vertices is referred as the $t$-vertex part of $K_{2,t}$ .*

**Definition 2.** *Let $\Gamma$ be a graph. The super subdivision of $\Gamma$ , denoted by $SSD_{(2,t)}(\Gamma)$ , is a new graph obtained from $\Gamma$ by replacing every edge $\lambda\mu$ of $\Gamma$ with a complete bipartite graph $K_{2,t}$ in such a way that the end vertices of each edge $\lambda\mu$ in $\Gamma$ are merged with the two vertices of two-vertex part of $K_{2,t}$ after removing the edge $\lambda\mu$ from $\Gamma$ .*

## 2. Complexity of the Super Subdivision Graph $SSD_{(2,t)}(C_n)$

**Lemma 4.** *Suppose $\mathcal{P}$ and $\mathcal{Q}$ are $\ell \times \ell$ block matrices, and $\Theta = \begin{pmatrix} \mathcal{P} & \mathcal{Q} & -\mathcal{Q} \\ \mathcal{Q} & \mathcal{P} & \mathcal{Q} \\ -\mathcal{Q} & \mathcal{Q} & \mathcal{P} \end{pmatrix}$.*

*Then, $det \, (\Theta) = [\, det(\, \mathcal{P} + \mathcal{Q})\,]^2 \, \times \, [\, det(\, \mathcal{P} - 2\mathcal{Q})\,]$.*

**Proof.** Utilising the characteristics of matrix row and column operations leads to

$$
det\,(\Theta) = det\ \begin{pmatrix} \mathcal{P} & \mathcal{Q} & -\mathcal{Q} \\ \mathcal{Q} & \mathcal{P} & \mathcal{Q} \\ -\mathcal{Q} & \mathcal{Q} & \mathcal{P} \end{pmatrix} \qquad adding\ C_2\ to\ C_1\ \ and\ C_2\ to\ \ C_3
$$

$$
= det\ \begin{pmatrix} \mathcal{P}+\mathcal{Q} & \mathcal{Q} & 0 \\ \mathcal{P}+\mathcal{Q} & \mathcal{P} & \mathcal{P}+\mathcal{Q} \\ 0 & \mathcal{Q} & \mathcal{P}+\mathcal{Q} \end{pmatrix} \qquad subtracting\ R_2\ from\ R_1
$$

$$
= det\ \begin{pmatrix} \mathcal{P}+\mathcal{Q} & \mathcal{Q} & 0 \\ 0 & \mathcal{P}-\mathcal{Q} & \mathcal{P}+\mathcal{Q} \\ 0 & \mathcal{Q} & \mathcal{P}+\mathcal{Q} \end{pmatrix} \qquad subtracting\ R_3\ from\ R_2
$$

$$
= det\ \begin{pmatrix} \mathcal{P}+\mathcal{Q} & \mathcal{Q} & 0 \\ 0 & \mathcal{P}-\mathcal{Q} & \mathcal{P}+\mathcal{Q} \\ 0 & 2\mathcal{Q}-\mathcal{P} & 0 \end{pmatrix} \qquad expanding\ along\ \ R_1
$$

$$
= [\,det\,(\,\mathcal{P}+\mathcal{Q})\,]^2 \times [\,det\,(\,\mathcal{P}-2\mathcal{Q})\,].\quad \square
$$

**Lemma 5.** *Suppose $\mathcal{P}$ and $\mathcal{Q}$ are $\ell \times \ell$ block matrices, and* $\mathrm{Y} = \begin{pmatrix} \mathcal{P} & \mathcal{Q} & \mathcal{Q} & \dots & \mathcal{Q} & \mathcal{Q} \\ \mathcal{Q} & \mathcal{P} & \mathcal{Q} & \dots & \mathcal{Q} & \mathcal{Q} \\ \mathcal{Q} & \mathcal{Q} & \mathcal{P} & \dots & \mathcal{Q} & \mathcal{Q} \\ \vdots & \dots & \dots & \ddots & \ddots & \vdots \\ \mathcal{Q} & \mathcal{Q} & \mathcal{Q} & \dots & \mathcal{P} & \mathcal{Q} \\ \mathcal{Q} & \mathcal{Q} & \mathcal{Q} & \dots & \mathcal{Q} & \mathcal{P} \end{pmatrix}_{k \times k}$

*Then, $det\,(\mathrm{Y}) = [\,det\,(\,\mathcal{P}-\mathcal{Q})\,]^{k-1} \times [\,det\,(\,\mathcal{P}+(k-1)\mathcal{Q})\,].$*

**Proof.** The results of employing the row and column characteristics of a matrix are as follows:

$$
det\,(\mathrm{Y}) = det\ \begin{pmatrix} \mathcal{P} & \mathcal{Q} & \mathcal{Q} & \dots & \mathcal{Q} & \mathcal{Q} \\ \mathcal{Q} & \mathcal{P} & \mathcal{Q} & \dots & \mathcal{Q} & \mathcal{Q} \\ \mathcal{Q} & \mathcal{Q} & \mathcal{P} & \dots & \mathcal{Q} & \mathcal{Q} \\ \vdots & \dots & \dots & \ddots & \ddots & \vdots \\ \mathcal{Q} & \mathcal{Q} & \mathcal{Q} & \dots & \mathcal{P} & \mathcal{Q} \\ \mathcal{Q} & \mathcal{Q} & \mathcal{Q} & \dots & \mathcal{Q} & \mathcal{P} \end{pmatrix}_{k \times k} \qquad subtracting\ C_k\ from\ all\ \ columns
$$

$$
= det\ \begin{pmatrix} \mathcal{P}-\mathcal{Q} & 0 & 0 & \dots & 0 & \mathcal{Q} \\ 0 & \mathcal{P}-\mathcal{Q} & 0 & \dots & 0 & \mathcal{Q} \\ 0 & \mathcal{Q}-\mathcal{P} & \mathcal{P}-\mathcal{Q} & \dots & 0 & \mathcal{Q} \\ \vdots & \dots & \dots & \ddots & \ddots & \vdots \\ 0 & 0 & 0 & \dots & \mathcal{P}-\mathcal{Q} & \mathcal{Q} \\ \mathcal{Q}-\mathcal{P} & 0 & 0 & \dots & \mathcal{Q}-\mathcal{P} & \mathcal{P} \end{pmatrix}_{k \times k} \qquad adding\ C_i\ to\ C_{i+1},\ 2 \le i \le k-1
$$

$$
= det\ \begin{pmatrix} \mathcal{P}-\mathcal{Q} & 0 & 0 & \dots & 0 & \mathcal{Q} \\ 0 & \mathcal{P}-\mathcal{Q} & 0 & \dots & 0 & \mathcal{Q} \\ 0 & 0 & \mathcal{P}-\mathcal{Q} & \dots & 0 & 2\mathcal{Q} \\ \vdots & \dots & \dots & \ddots & \ddots & \vdots \\ 0 & 0 & 0 & \dots & \mathcal{P}-\mathcal{Q} & (k-2)\mathcal{Q} \\ \mathcal{Q}-\mathcal{P} & 0 & 0 & \dots & 0 & \mathcal{P}+(k-2)\mathcal{Q} \end{pmatrix}_{k \times k} \qquad adding\ R_1\ to\ R_k
$$

$$
= det\ \begin{pmatrix} \mathcal{P}-\mathcal{Q} & 0 & 0 & \dots & 0 & \mathcal{Q} \\ 0 & \mathcal{P}-\mathcal{Q} & 0 & \dots & 0 & \mathcal{Q} \\ 0 & 0 & \mathcal{P}-\mathcal{Q} & \dots & 0 & 2\mathcal{Q} \\ \vdots & \dots & \ddots & \ddots & \ddots & \vdots \\ 0 & 0 & \ddots & \dots & \mathcal{P}-\mathcal{Q} & (k-2)\mathcal{Q} \\ 0 & 0 & \dots & \dots & 0 & \mathcal{P}+(k-1)\mathcal{Q} \end{pmatrix}_{k \times k}
$$

$$= [\, det\, (\, \mathcal{P} - \mathcal{Q})\,]^{k-1} \times [\, det\, (\, \mathcal{P} + (k-1)\mathcal{Q})\,]. \quad \square$$

### 2.1. Complexity of the Super Subdivision Graph $SSD_{(2,t)}(C_3)$

**Theorem 1.** *For any positive integer $t \geq 1$, the number of spanning trees of the super subdivision graph $SSD_{(2,t)}(C_3)$ of the cycle $C_3$ is given by:* $\quad \tau[SSD_{(2,t)}(C_3)] = 3\, t^2\, 2^{3t-2}.$

**Proof.** Let $C_3$ be a cycle with vertex set $\{u_k, 1 \leq k \leq 3\}$. The super subdivision graph $SSD_{(2,t)}(C_3)$ of $C_3$ has a vertex set $V[SSD_{(2,t)}(C_3)] = \{u_k, v_k^j, \ 1 \leq k \leq 3, 1 \leq j \leq t\}$. Thus, the graph $SSD_{(2,t)}(C_3)$ has $\alpha = |V[SSD_{(2,t)}(C_3)]| = 3(t+1)$ vertices and $\beta = 6t$ edges, see Figure 1.

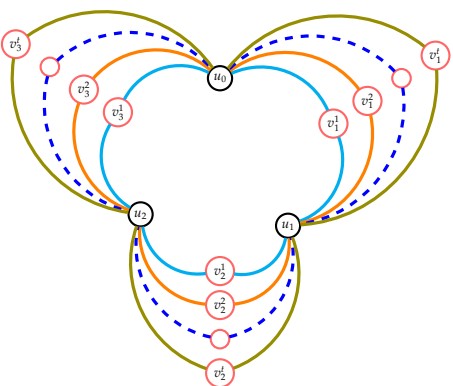

**Figure 1.** Super subdivision $SSD_{(2,t)}(C_3)$ of the cycle $C_3$.

Using Lemma 1, we get the following:

$$\tau[SSD_{(2,t)}(C_3)] = \frac{1}{(3t+3)^2}\, det\, [\, (3t+3)I - \overline{D} + \overline{A}\,] = \frac{1}{(3t+3)^2} \times$$

$$det \begin{pmatrix} 2t+1 & 1 & 1 & 0 & 0 & 0 & \ldots & 0 & 1 & 1 & \ldots & \ldots & 1 & 0 & 0 & 0 & \ldots & 0 \\ 1 & 2t+1 & 1 & 0 & 0 & 0 & \ldots & 0 & 0 & 0 & \ldots & \ldots & 0 & 1 & 1 & \ldots & \ldots & 1 \\ 1 & 1 & 2t+1 & 1 & 1 & \ldots & \ldots & 1 & 0 & 0 & 0 & \ldots & 0 & 0 & 0 & 0 & \ldots & 0 \\ 0 & 0 & 1 & 3 & 1 & 1 & \ldots & 1 & 1 & 1 & 1 & \ldots & 1 & 1 & 1 & \ldots & \ldots & 1 \\ 0 & 0 & 1 & 1 & 3 & 1 & \ldots & 1 & 1 & 1 & \ldots & \ldots & 1 & 1 & 1 & \ldots & \ldots & 1 \\ \vdots & \vdots & \vdots & \vdots & \ldots & \ddots & \ddots & \vdots & \vdots & \vdots & \ddots & \ddots & \vdots & \vdots & \vdots & \ddots & \ddots & \vdots \\ 0 & 0 & 1 & 1 & 1 & \ldots & 3 & 1 & 1 & 1 & \ldots & 1 & 1 & 1 & \ldots & \ldots & \ldots & 1 \\ 0 & 0 & 1 & 1 & \ldots & \ldots & 1 & 3 & 1 & 1 & \ldots & 1 & 1 & 1 & \ldots & \ldots & \ldots & 1 \\ 1 & 0 & 0 & 1 & 1 & \ldots & \ldots & 1 & 3 & 1 & 1 & \ldots & 1 & 1 & 1 & \ldots & \ldots & 1 \\ 1 & 0 & 0 & 1 & 1 & \ldots & \ldots & 1 & 1 & 3 & 1 & \ldots & 1 & 1 & 1 & \ldots & \ldots & 1 \\ \vdots & \vdots & \vdots & \vdots & \ldots & \ddots & \ddots & \vdots & \vdots & \vdots & \ddots & \ddots & \vdots & \vdots & \vdots & \ddots & \ddots & \vdots \\ 1 & 0 & 0 & 1 & 1 & \ldots & 1 & 1 & 1 & \ldots & \ldots & 3 & 1 & 1 & \ldots & \ldots & 1 & 1 \\ 1 & 0 & 0 & 1 & \ldots & \ldots & 1 & 1 & 1 & \ldots & \ldots & 1 & 3 & 1 & 1 & \ldots & \ldots & 1 \\ 0 & 1 & 0 & 1 & 1 & \ldots & \ldots & 1 & 1 & 1 & \ldots & \ldots & 1 & 3 & 1 & \ldots & \ldots & 1 \\ 0 & 1 & 0 & 1 & 1 & \ldots & \ldots & 1 & 1 & 1 & \ldots & \ldots & 1 & 1 & 3 & 1 & \ldots & 1 \\ \vdots & \vdots & \vdots & \vdots & \ldots & \ddots & \ddots & \vdots & \vdots & \vdots & \ddots & \ddots & \vdots & \vdots & \vdots & \ddots & \ddots & \vdots \\ 0 & 1 & 0 & 1 & 1 & \ldots & \ldots & 1 & 1 & 1 & \ldots & \ldots & 1 & 1 & \ldots & \ldots & 3 & 1 \\ 0 & 1 & 0 & 1 & 1 & \ldots & \ldots & 1 & 1 & 1 & \ldots & \ldots & 1 & 1 & \ldots & \ldots & 1 & 3 \end{pmatrix}_{(3t+3)\times(3t+3)}$$

$$= \frac{1}{(3t+3)^2}\, det\, \begin{pmatrix} \Phi_{(3\times3)} & \Delta_{(3\times3t)} \\ \Delta^t & \Psi_{(3t\times3t)} \end{pmatrix}$$

$$= \frac{1}{(3t+3)^2}[\, det(\Phi) \times det\, (\Psi - \Delta^t \Phi^{-1} \Delta)\,]. \tag{3}$$

$$det\ (\Phi) = det\ \begin{pmatrix} 2t+1 & 1 & 1 \\ 1 & 2t+1 & 1 \\ 1 & 1 & 2t+1 \end{pmatrix} = (2t+3)\ (2t)^2. \tag{4}$$

From Lemma 5, we have,

$$det\ (\Psi - \Delta^t\Phi^{-1}\Delta) = det\ \begin{pmatrix} \mathcal{P} & \mathcal{Q} & \mathcal{Q} \\ \mathcal{Q} & \mathcal{P} & \mathcal{Q} \\ \mathcal{Q} & \mathcal{Q} & \mathcal{P} \end{pmatrix} = [\ det\ (\mathcal{P} - \mathcal{Q})\ ]^2 \times det\ (\mathcal{P} + 2\mathcal{Q})\ ,$$

where $\mathcal{P} = \dfrac{1}{2t^2+3t} \begin{pmatrix} 6t^2+8t-1 & 2t^2+2t-1 & 2t^2+2t-1 & \dots & \dots & 2t^2+2t-1 \\ 2t^2+2t-1 & 6t^2+8t-1 & 2t^2+2t-1 & \dots & \dots & 2t^2+2t-1 \\ \vdots & \dots & \dots & \ddots & \ddots & \vdots \\ 2t^2+2t-1 & 2t^2+2t-1 & \dots & \dots & 6t^2+8t-1 & 2t^2+2t-1 \\ 2t^2+2t-1 & 2t^2+2t-1 & \dots & \dots & 2t^2+2t-1 & 6t^2+8t-1 \end{pmatrix}_{t \times t}$ and $\mathcal{Q} =$

$[\ a_{i,j}\ ]_{t \times t}$, is a matrix with the same value in all entries in which

$$a_{i,j} = \dfrac{4t^2+6t+1}{4t^2+6t}\ ,\ \text{then,}$$

$$det\ (\mathcal{P} - \mathcal{Q}) = (\dfrac{-1}{2t})^t\ det\ \begin{pmatrix} 1-4t & 1 & 1 & \dots & 1 \\ 1 & 1-4t & 1 & \dots & 1 \\ \vdots & \dots & \dots & \ddots & \vdots \\ 1 & \dots & \dots & 1-4t & 1 \\ 1 & \dots & \dots & 1 & 1-4t \end{pmatrix}_{t \times t}$$

$$= (\dfrac{-1}{2t})^t\ (-4t+t)\ (-4t)^{t-1} = 3\ 2^{t-2}. \tag{5}$$

Using matrix row and column operations and determinant characteristics, the following results are obtained:

$$det\ (\mathcal{P} + 2\mathcal{Q}) = (\dfrac{1}{2t+3})^t\ det\ \begin{pmatrix} 10t+14 & 6t+8 & 6t+8 & \dots & \dots & 6t+8 \\ 6t+8 & 10t+14 & 6t+8 & \dots & \dots & 6t+8 \\ \vdots & \vdots & \dots & \dots & \ddots & \vdots \\ 6t+8 & 6t+8 & \dots & \dots & 10t+14 & 6t+8 \\ 6t+8 & 6t+8 & \dots & \dots & 6t+8 & 10t+14 \end{pmatrix}_{t \times t}$$

$$= \dfrac{6(t+1)^2}{(2t+3)^t}\ det\ \begin{pmatrix} 1 & 0 & 0 & \dots & \dots & 0 \\ 1 & 4t+6 & 0 & \dots & \dots & 0 \\ \vdots & \vdots & \dots & \dots & \ddots & \vdots \\ 1 & 0 & \dots & \dots & 4t+6 & 0 \\ 1 & 0 & \dots & \dots & 0 & 4t+6 \end{pmatrix}_{t \times t}$$

$$= \dfrac{6(t+1)^2}{(2t+3)^t(4t+6)}\ det\ \begin{pmatrix} 4t+6 & 0 & 0 & \dots & \dots & 0 \\ 0 & 4t+6 & 0 & \dots & \dots & 0 \\ \vdots & \vdots & \dots & \dots & \ddots & \vdots \\ 0 & 0 & \dots & \dots & 4t+6 & 0 \\ 0 & 0 & \dots & \dots & 0 & 4t+6 \end{pmatrix}_{t \times t} = \dfrac{3\ 2^t\ (t+1)^2}{(2t+3)}. \tag{6}$$

Substituting Equations (4)–(6) in Equation (3), we obtain the result.  □

*2.2. Complexity of the Super Subdivision Graph $SSD_{(2,t)}(C_4)$*

**Theorem 2.** *For any positive integer $t \geq 1$, the number of spanning trees of the super subdivision graph $SSD_{(2,t)}(C_4)$ of the cycle $C_4$ is given by:*  $\tau[SSD_{(2,t)}(C_4)] = t^3\ 2^{4t-1}$.

**Proof.** Using the same approach as in Theorem 1, we have:

$$\tau[SSD_{(2,t)}(C_4)] = \frac{1}{(4t+4)^2} \, det \, \begin{pmatrix} \Phi_{(4\times4)} & \Delta_{(4\times4t)} \\ \Delta^t & \Psi_{(4t\times4t)} \end{pmatrix}$$

$$= \frac{1}{(4t+4)^2} [ \, det(\Phi) \, \times \, det \, (\Psi - \Delta^t\Phi^{-1}\Delta) \, ]. \tag{7}$$

$$det \, (\Phi) = det \, \begin{pmatrix} 2t+1 & 1 & 1 & 1 \\ 1 & 2t+1 & 1 & 1 \\ 1 & 1 & 2t+1 & 1 \\ 1 & 1 & 1 & 2t+1 \end{pmatrix} = (2t+4)\,(2t)^3. \tag{8}$$

From Lemma 2, we have,

$$det \, (\Psi - \Delta^t\Phi^{-1}\Delta) = det \, \begin{pmatrix} \mathcal{P} & \mathcal{Q} & \mathcal{R} & \mathcal{Q} \\ \mathcal{Q} & \mathcal{P} & \mathcal{Q} & \mathcal{R} \\ \mathcal{R} & \mathcal{Q} & \mathcal{P} & \mathcal{Q} \\ \mathcal{Q} & \mathcal{R} & \mathcal{Q} & \mathcal{P} \end{pmatrix},$$

where $\mathcal{P} = \dfrac{1}{(t^2+2t)^t} \begin{pmatrix} 3t^2+5t-1 & t^2+t-1 & t^2+t-1 & \dots & \dots & t^2+t-1 \\ t^2+t-1 & 3t^2+5t-1 & t^2+t-1 & \dots & \dots & t^2+t-1 \\ \vdots & \dots & \dots & \ddots & \ddots & \vdots \\ t^2+t-1 & t^2+t-1 & \dots & \dots & 3t^2+5t-1 & t^2+t-1 \\ t^2+t-1 & t^2+t-1 & \dots & \dots & t^2+t-1 & 3t^2+5t-1 \end{pmatrix}_{t\times t}$, $\mathcal{R} =$

$[\,a_{ij}\,]_{t\times t}$, in which $a_{ij} = \dfrac{t^2+2t+1}{t^2+2t}$, and $\mathcal{Q} = [\,b_{ij}\,]_{t\times t}$, in which $b_{ij} = \dfrac{2t+3}{2t+4}$.

Performing $\mathcal{H} = det \, \begin{pmatrix} \mathcal{P} & \mathcal{Q} \\ \mathcal{Q} & \mathcal{P} \end{pmatrix}$ and $\mathcal{G} = det \, \begin{pmatrix} \mathcal{R} & \mathcal{Q} \\ \mathcal{Q} & \mathcal{R} \end{pmatrix}$, we have

$$det \, (\Psi - \Delta^t\Phi^{-1}\Delta) = det \, \begin{pmatrix} \mathcal{H} & \mathcal{G} \\ \mathcal{H} & \mathcal{G} \end{pmatrix} = [ \, det(\mathcal{H}+\mathcal{G}) \, \times \, det(\mathcal{H}-\mathcal{G}) \, ]$$

$$= det \, \begin{pmatrix} \mathcal{P}+\mathcal{R} & 2\mathcal{Q} \\ 2\mathcal{Q} & \mathcal{P}+\mathcal{R} \end{pmatrix} \times det \, \begin{pmatrix} \mathcal{P}-\mathcal{R} & \mathcal{O} \\ \mathcal{O} & \mathcal{P}-\mathcal{R} \end{pmatrix}$$

$$= det(\mathcal{P}+\mathcal{R}+2\mathcal{Q}) \, \times \, det \, (\mathcal{P}+\mathcal{R}-2\mathcal{Q}) \, \times \, [det \, (\mathcal{P}-\mathcal{R})]^2.$$

$$det \, (\mathcal{P}+\mathcal{R}+2\mathcal{Q}) = (\frac{1}{t+2})^t det \, \begin{pmatrix} 6t+10 & 4t+6 & \dots & \dots & 4t+6 \\ 4t+6 & 6t+10 & \dots & \dots & 4t+6 \\ \vdots & \vdots & \dots & \ddots & \vdots \\ 4t+6 & \dots & \dots & 6t+10 & 4t+6 \\ 4t+6 & \dots & \dots & 4t+6 & 6t+10 \end{pmatrix}_{t\times t}$$

$$= \frac{4t^2+8t+4}{(t+2)^t(2t+4)} det \, \begin{pmatrix} 2t+4 & 0 & \dots & \dots & 0 \\ 0 & 2t+4 & 0 & \dots & 0 \\ \vdots & \vdots & \dots & \ddots & \vdots \\ 0 & \dots & \dots & 2t+4 & 0 \\ 0 & \dots & \dots & 0 & 2t+4 \end{pmatrix}_{t\times t} = \frac{2^{t+1}\,(t+1)^2}{(t+2)}. \tag{9}$$

$$det(\mathcal{P}+\mathcal{R}-2\mathcal{Q}) = det \, \begin{pmatrix} 2 & 0 & 0 & \dots & \dots & 0 \\ 0 & 2 & 0 & \dots & \dots & 0 \\ \vdots & \vdots & \dots & \dots & \ddots & \vdots \\ 0 & 0 & \dots & \dots & 2 & 0 \\ 0 & 0 & \dots & \dots & 0 & 2 \end{pmatrix}_{t\times t} = 2^t. \tag{10}$$

Lemma 3 provides us with,

$$det\,(\mathcal{P}-\mathcal{R}) = (\frac{-1}{t})^t\,det\,\begin{pmatrix} 1-2t & 1 & \cdots & \cdots & 1 \\ 1 & 1-2t & 1 & \cdots & 1 \\ \vdots & \cdots & \cdots & \ddots & \vdots \\ 1 & \cdots & \cdots & 1-2t & 1 \\ 1 & \cdots & \cdots & 1 & 1-2t \end{pmatrix}_{t\times t} = 2^{t-1}. \qquad (11)$$

Substituting Equations (8)–(11) in Equation (7), we obtain the result. □

### 2.3. Complexity of the Super Subdivision Graph $SSD_{(2,t)}(C_5)$

**Theorem 3.** *For any positive integer $t \geq 1$, the number of spanning trees of the super subdivision graph $SSD_{(2,t)}[C_5]$ of the cycle $C_5$ is given by:* $\tau[SSD_{(2,t)}(C_5)] = 5\,t^4\,2^{5t-4}$.

**Proof.** Using the same technique as in Theorem 1, we have:

$$\tau[SSD_{(2,t)}(C_5)] = \frac{1}{(5t+5)^2}[\,det(\Phi_{(5\times5)}) \times det\,(\Psi_{(5t\times5t)} - \Delta^t\Phi^{-1}\Delta_{(5\times5t)})\,]. \qquad (12)$$

$$det\,(\Phi_{(5\times5)}) = (2t+5)\,(2t)^4. \qquad (13)$$

$$det\,(\Psi - \Delta^t\Phi^{-1}\Delta) = det\,\begin{pmatrix} \mathcal{P} & \mathcal{Q} & \mathcal{R} & \mathcal{R} & \mathcal{Q} \\ \mathcal{Q} & \mathcal{P} & \mathcal{Q} & \mathcal{R} & \mathcal{R} \\ \mathcal{R} & \mathcal{Q} & \mathcal{P} & \mathcal{Q} & \mathcal{R} \\ \mathcal{R} & \mathcal{R} & \mathcal{Q} & \mathcal{P} & \mathcal{Q} \\ \mathcal{Q} & \mathcal{R} & \mathcal{R} & \mathcal{Q} & \mathcal{P} \end{pmatrix}$$

$$= det\,\begin{pmatrix} \mathcal{P}+2\mathcal{Q}+2\mathcal{R} & 2\mathcal{Q} & 2\mathcal{Q}+2\mathcal{R} & \mathcal{R} & \mathcal{Q} \\ 0 & \mathcal{Q}-\mathcal{P} & \mathcal{Q}-\mathcal{R} & \mathcal{Q}-\mathcal{R} & 0 \\ 0 & \mathcal{Q}-\mathcal{R} & \mathcal{R}-\mathcal{P} & \mathcal{R}-\mathcal{Q} & \mathcal{Q}-\mathcal{R} \\ 0 & 0 & 0 & \mathcal{Q}-\mathcal{P} & \mathcal{R}-\mathcal{Q} \\ 0 & 0 & 0 & \mathcal{R}-\mathcal{Q} & \mathcal{R}-\mathcal{P} \end{pmatrix}$$

$$= det\,(\mathcal{P}+2\mathcal{Q}+2\mathcal{R})\,det\,\begin{pmatrix} \mathcal{Q}-\mathcal{P} & \mathcal{Q}-\mathcal{R} & 0 & 0 \\ \mathcal{Q}-\mathcal{R} & \mathcal{R}-\mathcal{P} & 0 & 0 \\ 0 & 0 & \mathcal{Q}-\mathcal{P} & \mathcal{R}-\mathcal{Q} \\ 0 & 0 & \mathcal{R}-\mathcal{Q} & \mathcal{R}-\mathcal{P} \end{pmatrix}$$

$$= det\,(\mathcal{P}+2\mathcal{Q}+2\mathcal{R})\,det\,[(\mathcal{Q}-\mathcal{P})(\mathcal{R}-\mathcal{P}) - (\mathcal{Q}-\mathcal{R})^2]^2$$

where $\mathcal{P} = \dfrac{1}{(2t^2+5t)^t}\begin{pmatrix} 6t^2+12t-3 & 2t^2+2t-3 & 2t^2+2t-3 & \cdots & \cdots & 2t^2+2t-3 \\ 2t^2+2t-3 & 6t^2+12t-3 & 2t^2+2t-3 & \cdots & \cdots & 2t^2+2t-3 \\ \vdots & \cdots & \cdots & \ddots & \ddots & \vdots \\ 2t^2+2t-3 & 2t^2+2t-3 & \cdots & \cdots & 6t^2+12t-3 & 2t^2+2t-3 \\ 2t^2+2t-3 & 2t^2+2t-3 & \cdots & \cdots & 2t^2+2t-3 & 6t^2+12t-3 \end{pmatrix}_{t\times t}$,

$\mathcal{Q} = [a_{ij}]_{t\times t}$, in which $a_{ij} = \dfrac{4t^2+6t-1}{4t^2+10t}$, and $\mathcal{R} = [b_{ij}]_{t\times t}$, in which $b_{ij} = \dfrac{2t^2+4t+2}{2t^2+5t}$.

$$det\,(\mathcal{P}+2\mathcal{Q}+2\mathcal{R}) = \frac{5\,2^t\,(t+1)^2}{(2t+5)}. \qquad (14)$$

A straightforward calculation reveals that:

$$(\mathcal{Q}-\mathcal{P})(\mathcal{R}-\mathcal{P}) = (\frac{1}{2t})^t\begin{pmatrix} 8t-5 & -5 & \cdots & \cdots & -5 \\ -5 & 8t-5 & \cdots & \cdots & -5 \\ \vdots & \cdots & \cdots & \ddots & \vdots \\ -5 & \cdots & \cdots & 8t-5 & -5 \\ -5 & \cdots & \cdots & -5 & 8t-5 \end{pmatrix}_{t\times t}$$

and $(Q - R) = [c_{ij}]_{t \times t}$, in which $c_{ij} = \dfrac{-1}{2t}$, then $(Q - R)^2 = [h_{ij}]_{t \times t}$, in which $h_{ij} = \dfrac{1}{4t}$.

$$det\,[(Q - P)(R - P) - (Q - R)^2] = 5\,2^{2t-4}. \tag{15}$$

Substituting Equations (13)–(15) in Equation (12), we obtain the result. $\square$

*2.4. Complexity of the Super Subdivision Graph $SSD_{(2,t)}(C_6)$*

**Theorem 4.** *For any positive integer $t \geq 1$, the number of spanning trees of the super subdivision graph $SSD_{(2,t)}(C_6)$ of the cycle $C_6$ is given by:* $\tau[SSD_{(2,t)}(C_6)] = 3\,t^5\,2^{6t-4}.$

**Proof.** Using the same approach as in Theorem 1, we have:

$$\tau[SSD_{(2,t)}(C_6)] = \frac{1}{(6t+6)^2}\,[\,det(\Phi_{(6\times6)}) \times det\,(\Psi_{(6t\times6t)} - \Delta^t\Phi^{-1}\Delta_{(6\times6t)})\,]. \tag{16}$$

$$det\,(\Phi_{(6\times6)}) = (2t+6)\,(2t)^5. \tag{17}$$

Applying Lemma 4 and Lemma 5, we have

$$det(\Psi - \Delta^t\Phi^{-1}\Delta) = det \begin{pmatrix} P & Q & R & R & R & Q \\ Q & P & Q & R & R & R \\ R & Q & P & Q & R & R \\ R & R & Q & P & Q & R \\ R & R & R & Q & P & Q \\ Q & R & R & R & Q & P \end{pmatrix} = det \begin{pmatrix} H & G \\ G & H \end{pmatrix} = det(H + G) \times det(H - G)$$

$$= det \begin{pmatrix} P+R & Q+R & Q+R \\ Q+R & P+R & Q+R \\ Q+R & Q+R & P+R \end{pmatrix} \times det \begin{pmatrix} P-R & Q-R & R-Q \\ Q-R & P-R & Q-R \\ R-Q & Q-R & P-R \end{pmatrix}$$

$$= [det\,(P - Q)]^2 \times det\,(P + 2Q + 3R) \times [det\,(P + Q - 2R)]^2 \times$$
$$det(P + R - 2Q).$$

where $P = \dfrac{1}{(t^2+3t)^t} \begin{pmatrix} 3t^2+7t-2 & t^2+t-2 & t^2+t-2 & \ldots & \ldots & t^2+t-2 \\ t^2+t-2 & 3t^2+7t-2 & t^2+t-2 & \ldots & \ldots & t^2+t-2 \\ \vdots & & \ddots & \ddots & \ddots & \vdots \\ t^2+t-2 & t^2+t-2 & \ldots & \ldots & 3t^2+7t-2 & t^2+t-2 \\ t^2+t-2 & t^2+t-2 & \ldots & \ldots & t^2+t-2 & 3t^2+7t-2 \end{pmatrix}_{t \times t}$,

$Q = [a_{ij}]_{t \times t}$, in which $a_{ij} = \dfrac{2t^2+3t-1}{2t^2+6t}$ and $R = [b_{ij}]_{t \times t}$, in which $b_{ij} = \dfrac{t^2+2t+1}{t^2+3t}$.

$$det\,(P - Q) = 3\,2^{t-2}. \tag{18}$$

$$det\,(P + 2Q + 3R) = \frac{3\,2^t\,(t+1)^2}{(t+3)}. \tag{19}$$

$$det\,(P + Q - 2R) = 2^{t-2}. \tag{20}$$

$$det\,(P + R - 2Q) = 2^t. \tag{21}$$

From substituting Equations (17)–(21), in (16), we obtain the result. $\square$

*2.5. Application*

2.5.1. The Dumbbell Graph $Db_{m,n}$

**Theorem 5.** *For any positive integer $t \geq 1$, the number of spanning trees of the super subdivision graph $SSD_{(2,t)}(K_2)$ of the complete graph $K_2$ is given by:* $\tau[SSD_{(2,t)}(K_2)] = t\,2^{t-1}.$

**Proof.** Applying Lemma 1, we obtain:

$$\tau[SSD_{(2,t)}(K_2)] = \frac{1}{(t+2)^2} \det \begin{pmatrix} t+1 & 1 & 0 & 0 & \cdots & \cdots & 0 \\ 1 & t+1 & 0 & 0 & \cdots & \cdots & 0 \\ 0 & 0 & 3 & 1 & \cdots & \cdots & 1 \\ 0 & 0 & 1 & 3 & 1 & \cdots & 1 \\ \vdots & \vdots & \cdots & \cdots & \cdots & \ddots & \vdots \\ 0 & 0 & 1 & 1 & \cdots & 1 & 3 \end{pmatrix}_{t \times t} = \frac{1}{(t+2)^2} \det \begin{pmatrix} \mathcal{P} & O \\ O & \mathcal{Q} \end{pmatrix}$$

$$= \frac{1}{(t+2)^2} \det(\mathcal{P}) \times \det(\mathcal{Q}) = \frac{1}{(t+2)^2}(t^2 + 2t)(3 + t - 1)(3 - 1)^{t-1} = t\, 2^{t-1}. \quad \square$$

**Definition 3.** *The graph created by joining two disjoint cycles* $C_m = \{u_1, u_2, \cdots, u_m\}$ *and* $C_n = \{v_1, v_2, \cdots, v_n\}$ *with an edge* $r = u_1 v_n$ *is called* a dumbbell graph *, represented by* $Db_{m,n}$. *The super subdivision* $SSD_{(2,t)}(Db_{4,6})$ *of the dumbbell graph* $Db_{4,6}$ *is represented in Figure 2.*

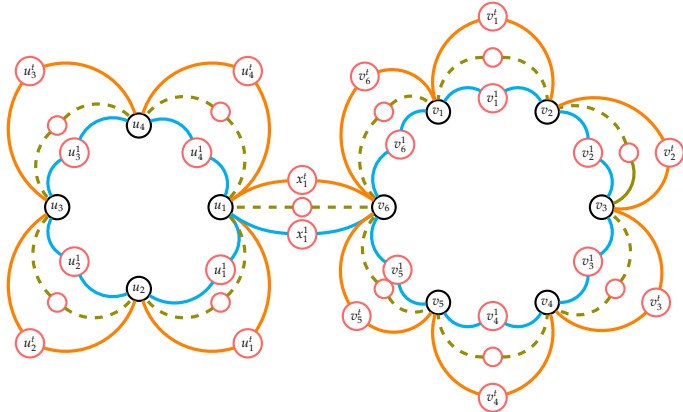

**Figure 2.** Super subdivision $SSD_{(2,t)}(Db_{4,6})$ of the dumbbell graph $Db_{4,6}$.

**Definition 4** ([13]). *By* $G_1 \oplus_v G_2$, *we mean the union of two graphs* $G_1$ *and* $G_2$ *that share a single vertex, vs., on their common external face.*

Using Theorem (2.4) in [13] which states that if the graphs $G_1, G_2, \cdots, G_n$ have a common vertex in the external face, then $\tau(G_1 \oplus_v G_2 \oplus_v \cdots \oplus_v G_n) = \tau(G_1) \times \tau(G_2) \times \cdots \times \tau(G_n)$, then we have the following theorem:

**Theorem 6.** *For any positive integer* $t \geq 1$, *then*

| | |
|---|---|
| *(i)* $\tau[SSD_{(2,t)}(Db_{3,3})] = 9\, t^5\, 2^{7t-5}$. | *(ii)* $\tau[SSD_{(2,t)}(Db_{3,4})] = 3\, t^6\, 2^{8t-4}$. |
| *(iii)* $\tau[SSD_{(2,t)}(Db_{3,5})] = 15\, t^7\, 2^{9t-7}$. | *(iv)* $\tau[SSD_{(2,t)}(Db_{3,6})] = 9\, t^8\, 2^{10t-7}$. |
| *(v)* $\tau[SSD_{(2,t)}(Db_{4,4})] = t^7\, 2^{9t-3}$. | *(vi)* $\tau[SSD_{(2,t)}(Db_{4,5})] = 5\, t^8\, 2^{10t-6}$. |
| *(vii)* $\tau[SSD_{(2,t)}(Db_{4,6})] = 3\, t^9\, 2^{11t-6}$. | *(viii)* $\tau[SSD_{(2,t)}(Db_{5,5})] = 25\, t^9\, 2^{11t-9}$. |
| *(ix)* $\tau[SSD_{(2,t)}(Db_{5,6})] = 15\, t^{10}\, 2^{12t-9}$. | *(x)* $\tau[SSD_{(2,t)}(Db_{6,6})] = 9\, t^{11}\, 2^{13t-9}$. |

**Proof.** Using Theorems 1–5, we obtain:

(i) $\quad SSD_{(2,t)}(Db_{3,3}) = SSD_{(2,t)}(C_3) \oplus_v SSD_{(2,t)}(K_2) \oplus_v SSD_{(2,t)}(C_3).$

$$\tau[SSD_{(2,t)}(Db_{3,3})] = \tau[SSD_{(2,t)}(C_3)] \times \tau[SSD_{(2,t)}(K_2)] \times \tau[SSD_{(2,t)}(C_3)]$$
$$= [3\, t^2\, 2^{3t-2}] \times [t\, 2^{t-1}] \times [3\, t^2\, 2^{3t-2}] = 9\, t^5\, 2^{7t-5}.$$

(ii) $\quad SSD_{(2,t)}(Db_{3,4}) = SSD_{(2,t)}(C_3) \oplus_v SSD_{(2,t)}(K_2) \oplus_v SSD_{(2,t)}(C_4).$

$$\tau[SSD_{(2,t)}(Db_{3,4})] = \tau[SSD_{(2,t)}(C_3)] \times \tau[SSD_{(2,t)}(K_2)] \times \tau[SSD_{(2,t)}(C_4)]$$
$$= [3\, t^2\, 2^{3t-2}] \times [t\, 2^{t-1}] \times [t^3\, 2^{4t-1}] = 3\, t^6\, 2^{8t-4}.$$

The proofs of (iii–x) follow similarly, as in (i–ii). □

### 2.5.2. The Dragon Graph $P_m(C_n)$

*The dragon graph*, represented by $P_m(C_n)$, is created by matching vertex $v_m$ of the path $P_m$ with vertex $u_1$ of a cycle graph $C_n$. The vertex set of $P_m(C_n)$ is $\{v_k, \ 1 \leq k \leq m-1\} \cup \{u_k, \ 1 \leq k \leq n\}$. The super subdivision graph $SSD_{(2,t)}[P_m(C_n)]$ of the dragon graph has a vertex set $V[SSD_{(2,t)}(P_m(C_n))] = \{v_k, v_k^j, \ k \in [1, m-1]\} \cup \{u_k, u_k^j, \ k \in [1, n]\}$. As a result, the graph $SSD_{(2,t)}[P_m(C_n)]$ has $\alpha = (m+n-1)(t+1)$ vertices and $\beta = 2t(m+n-1)$ edges, see Figure 3.

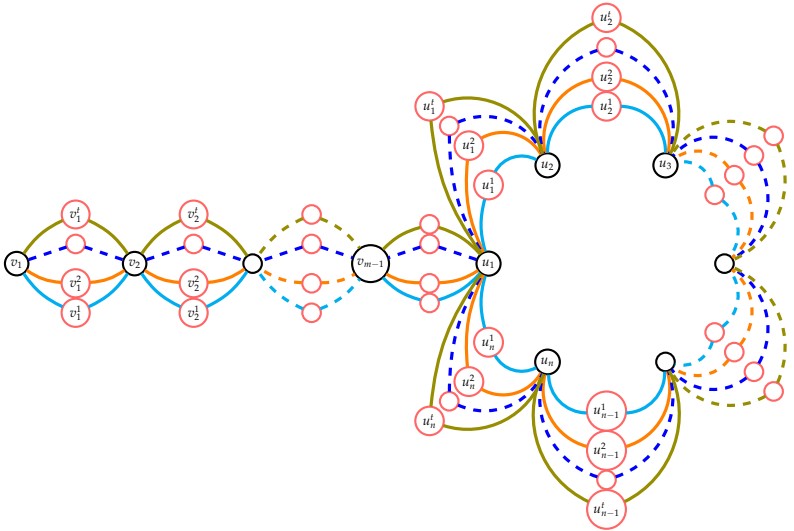

**Figure 3.** Super subdivision $SSD_{(2,t)}[P_m(C_n)]$ of the dragon graph $P_m(C_n)$.

**Theorem 7.** *For any positive integer $t \geq 1$, the number of spanning trees of the super subdivision graph $SSD_{(2,t)}(P_n)$ of the path is given by:* $\tau[SSD_{(2,t)}(P_n)] = [t\, 2^{t-1}]^{n-1}$.

**Proof.** Let $P_n$ be the path with vertex set $\{u_k, \ k \in [1, n]\}$. The super subdivision graph $SSD_{(2,t)}[P_n]$ of the path $P_n$ has a vertex set $V[SSD_{(2,t)}(P_n)] = \{u_k, \ k \in [1, n]\} \cup \{v_k^j, \ k \in [1, n-1]\}$. As a result, the graph $SSD_{(2,t)}(P_n)$ has $\alpha = n + t(n-1)$ vertices and $\beta = 2t(n-1)$ edges. Using Theorem 5, we obtain:

$$SSD_{(2,t)}(P_n) = \underbrace{SSD_{(2,t)}(K_2) \oplus_v SSD_{(2,t)}(K_2) \oplus_v \cdots \oplus_v SSD_{(2,t)}(K_2)}_{(n-1) \text{ times.}}$$

$$\tau[SSD_{(2,t)}(P_n)] = \underbrace{[\tau(SSD_{(2,t)}(K_2)] \times [\tau(SSD_{(2,t)}(K_2)] \times \cdots \times [\tau(SSD_{(2,t)}(K_2))]}_{(n-1) \text{ times.}}$$

$$= [\tau(SSD_{(2,t)}(K_2))]^{n-1} = [t\, 2^{t-1}]^{n-1} = t^{n-1}\, 2^{(n-1)(t-1)}. \quad \square$$

**Theorem 8.** *For any positive integer $t \geq 1$, then*

| (i) $\tau(SSD_{(2,t)}[P_m(C_3)]) = 3\, t^{m+1}\, 2^{m(t-1)+2t-1}$. | (ii) $\tau(SSD_{(2,t)}[P_m(C_4)]) = t^{m+2}\, 2^{m(t-1)+3t}$. |
|---|---|
| (iii) $\tau(SSD_{(2,t)}[P_m(C_5)]) = 5\, t^{m+3}\, 2^{m(t-1)+4t-3}$. | (iv) $\tau(SSD_{(2,t)}[P_m(C_6)]) = 3\, t^{m+4}\, 2^{m(t-1)+5t-3}$. |

**Proof.** Using Theorems 1–4 and Theorem 7, we obtain:

(i)  Since  $SSD_{(2,t)}[P_m(C_3)] = SSD_{(2,t)}(C_3) \oplus_v SSD_{(2,t)}(P_m)$, then

$$\begin{aligned}
\tau(SSD_{(2,t)}[P_m(C_3)]) &= \tau[SSD_{(2,t)}(C_3)] \times \tau[SSD_{(2,t)}(P_m)] \\
&= [3\, t^2\, 2^{3t-2}] \times [(t\, 2^{t-1})]^{m-1} = 3\, t^{m+1}\, 2^{m(t-1)+2t-1}.
\end{aligned}$$

(ii)  Since  $SSD_{(2,t)}[P_m(C_4)] = SSD_{(2,t)}(C_4) \oplus_v SSD_{(2,t)}(P_m)$, then

$$\begin{aligned}
\tau(SSD_{(2,t)}[P_m(C_4)]) &= \tau[SSD_{(2,t)}(C_4)] \times \tau[SSD_{(2,t)}(P_m)] \\
&= [t^3\, 2^{4t-1}] \times [(t\, 2^{t-1})]^{m-1} = t^{m+2}\, 2^{m(t-1)+3t}.
\end{aligned}$$

The proofs of (iii–iv) follow similarly, as in (i–ii).  □

### 3. Complexity of the Super Subdivision Graph $SSD_{(2,t)}(\Pi_n)$ of the Prism $\Pi_n$

Let $C_n = \{v_1, v_2, \cdots, v_n\}$ be a cycle and $C'_n = \{u_1, u_2, \cdots, u_n\}$ be a copy of $C_n$. The prism $\prod_n$ is constructed by joining each vertex $v_i$ of $C_n$ to the corresponding vertex $u_i$ of $C'_n$ for all $i \in \{1, 2, 3, \cdots, n\}$, the edge set is $E(\Pi_n) = \{v_i v_{i+1}, v_i u_i, u_i u_{i+1}, , i = 1, 2, \cdots, n\}$. Thus, $\alpha(\Pi_n) = 2n$ and $\beta(\Pi_n) = 3n$.

**Lemma 6.** *Suppose $\mathcal{P}$, $\mathcal{Q}$ and $\mathcal{R}$ are $t \times t$ block matrices and $\Omega =$*
$$\begin{pmatrix}
\mathcal{P} & \mathcal{Q} & \mathcal{Q} & \mathcal{R} & \mathcal{R} & \mathcal{R} & \mathcal{Q} & \mathcal{Q} & \mathcal{R} \\
\mathcal{Q} & \mathcal{P} & \mathcal{Q} & \mathcal{R} & \mathcal{R} & \mathcal{R} & \mathcal{R} & \mathcal{Q} & \mathcal{Q} \\
\mathcal{Q} & \mathcal{Q} & \mathcal{P} & \mathcal{R} & \mathcal{R} & \mathcal{R} & \mathcal{Q} & R & \mathcal{Q} \\
\mathcal{R} & \mathcal{R} & \mathcal{R} & \mathcal{P} & \mathcal{Q} & \mathcal{Q} & \mathcal{Q} & \mathcal{Q} & \mathcal{R} \\
\mathcal{R} & \mathcal{R} & \mathcal{R} & \mathcal{Q} & \mathcal{P} & \mathcal{Q} & \mathcal{R} & \mathcal{Q} & \mathcal{Q} \\
\mathcal{R} & \mathcal{R} & \mathcal{R} & \mathcal{Q} & \mathcal{Q} & \mathcal{P} & \mathcal{Q} & \mathcal{R} & \mathcal{Q} \\
\mathcal{Q} & \mathcal{R} & \mathcal{Q} & \mathcal{Q} & \mathcal{R} & \mathcal{Q} & \mathcal{P} & \mathcal{R} & \mathcal{R} \\
\mathcal{Q} & \mathcal{Q} & \mathcal{R} & \mathcal{Q} & \mathcal{Q} & \mathcal{R} & \mathcal{R} & \mathcal{P} & \mathcal{R} \\
\mathcal{R} & \mathcal{Q} & \mathcal{Q} & \mathcal{R} & \mathcal{Q} & \mathcal{Q} & \mathcal{R} & \mathcal{R} & \mathcal{P}
\end{pmatrix}_{9 \times 9}$$

*Then, $\det \Omega = [\det (\mathcal{P} - \mathcal{Q})]^2 \det (3\mathcal{R} - 2\mathcal{Q} - \mathcal{P}) [\det (2\mathcal{Q} - \mathcal{P} - \mathcal{R})]^2 [\det (2\mathcal{R} - \mathcal{P} - \mathcal{Q})]^2 \times$*
$$\det [(2\mathcal{Q} + \mathcal{R})(4\mathcal{Q} - \mathcal{P}) - (2\mathcal{R} + \mathcal{P})^2]$$

**Proof.** The row and column properties of matrices are applied to the matrix $\Omega$ and the following operations are performed consecutively:

(1)  Subtracting $R_4$ from $R_1$, $R_5$ from $R_2$, and $R_6$ from $R_3$;
(2)  Adding $C_1$ to $C_4$, $C_2$ to $C_5$, and $C_3$ to $C_6$;
(3)  Subtracting $R_2$ from $R_1$;
(4)  Adding $C_1$ to $C_2$;
(5)  Expanding along $R_1$;
(6)  Adding $C_1$ to $C_2$;
(7)  Subtracting $R_2$ from $R_1$.

Then, we obtain

$$\det \Omega = \det (\mathcal{P} - \mathcal{Q}) \det
\begin{pmatrix}
\mathcal{P} - \mathcal{Q} & 0 & 0 & 0 & 0 & 0 & 0 & 0 \\
2\mathcal{R} - 2\mathcal{Q} & 3\mathcal{R} - 2\mathcal{Q} - \mathcal{P} & 0 & 0 & 0 & 0 & 0 & 0 \\
2\mathcal{R} & 3\mathcal{R} & \mathcal{R} + \mathcal{P} & \mathcal{R} + \mathcal{Q} & \mathcal{R} + \mathcal{Q} & \mathcal{Q} & \mathcal{Q} & \mathcal{R} \\
2\mathcal{R} & 3\mathcal{R} & \mathcal{R} + \mathcal{Q} & \mathcal{R} + \mathcal{P} & \mathcal{R} + \mathcal{Q} & \mathcal{R} & \mathcal{Q} & \mathcal{Q} \\
2\mathcal{R} & 3\mathcal{R} & \mathcal{R} + \mathcal{Q} & \mathcal{R} + \mathcal{Q} & \mathcal{R} + \mathcal{P} & \mathcal{Q} & \mathcal{R} & \mathcal{Q} \\
\mathcal{Q} + \mathcal{R} & 2\mathcal{Q} + \mathcal{R} & 2\mathcal{Q} & 2\mathcal{R} & 2\mathcal{Q} & \mathcal{P} & \mathcal{R} & \mathcal{R} \\
2\mathcal{Q} & 2\mathcal{Q} + \mathcal{R} & 2\mathcal{Q} & 2\mathcal{Q} & 2\mathcal{R} & \mathcal{R} & \mathcal{P} & \mathcal{R} \\
\mathcal{R} + \mathcal{Q} & 2\mathcal{Q} + \mathcal{R} & 2\mathcal{R} & 2\mathcal{Q} & 2\mathcal{Q} & \mathcal{R} & \mathcal{R} & \mathcal{P}
\end{pmatrix}_{8 \times 8}$$

$$= [\det (\mathcal{P} - \mathcal{Q})]^2 \det (3\mathcal{R} - 2\mathcal{Q} - \mathcal{P}) \det
\begin{pmatrix}
\mathcal{R} + \mathcal{P} & \mathcal{R} + \mathcal{Q} & \mathcal{R} + \mathcal{Q} & \mathcal{Q} & \mathcal{Q} & \mathcal{R} \\
\mathcal{R} + \mathcal{Q} & \mathcal{R} + \mathcal{P} & \mathcal{R} + \mathcal{Q} & \mathcal{R} & \mathcal{Q} & \mathcal{Q} \\
\mathcal{R} + \mathcal{Q} & \mathcal{R} + \mathcal{Q} & \mathcal{R} + \mathcal{P} & \mathcal{Q} & \mathcal{R} & \mathcal{Q} \\
2\mathcal{Q} & 2\mathcal{R} & 2\mathcal{Q} & \mathcal{P} & \mathcal{R} & \mathcal{R} \\
2\mathcal{Q} & 2\mathcal{Q} & 2\mathcal{R} & \mathcal{R} & \mathcal{P} & \mathcal{R} \\
2\mathcal{R} & 2\mathcal{Q} & 2\mathcal{Q} & \mathcal{R} & \mathcal{R} & \mathcal{P}
\end{pmatrix}_{6 \times 6}$$

The following operations are performed consecutively on the above matrix:

(1) Subtracting $R_2$ from $R_3$ and $R_4$ from $R_5$;
(2) Adding $C_3$ to $C_2$ and $C_5$ to $C_4$;
(3) Subtracting $R_6$ from $R_4$;
(4) Subtracting $C_4$ from $C_1$;
(5) Subtracting $R_1$ from $R_4$;
(6) Expanding along $C_1$;
(7) Subtracting $C_2$ from $C_2$ and $C_5$ from $C_4$;
(8) Adding $C_2$ to $C_4$;
(9) Subtracting $R_4$ from $R_2$ .

Then, we obtain

$$det\ \Omega = [det(\mathcal{P} - \mathcal{Q})]^2\ det\ (3\mathcal{R} - 2\mathcal{Q} - \mathcal{P})\ det\ (2\mathcal{Q} - \mathcal{R} - \mathcal{P}) \times$$

$$det\ \begin{pmatrix} \mathcal{P} + \mathcal{Q} + 2\mathcal{R} & 0 & \mathcal{R} + \mathcal{Q} & 0 & \mathcal{Q} \\ 0 & 2\mathcal{R} - \mathcal{Q} - \mathcal{P} & 0 & 0 & 0 \\ 4\mathcal{R} & \mathcal{P} - 2\mathcal{R} + \mathcal{Q} & 2\mathcal{Q} - \mathcal{R} + \mathcal{P} & 0 & 2\mathcal{R} - \mathcal{P} \\ 0 & 2\mathcal{R} - 2\mathcal{Q} & 0 & \mathcal{P} - 2\mathcal{Q} + \mathcal{R} & 0 \\ 4\mathcal{Q} & 2\mathcal{R} - 2\mathcal{Q} & 2\mathcal{R} & \mathcal{P} - 2\mathcal{Q} + \mathcal{R} & \mathcal{P} \end{pmatrix}_{5 \times 5}$$

$$det\ \Omega = [det(\mathcal{P} - \mathcal{Q})]^2\ det(3\mathcal{R} - 2\mathcal{Q} - \mathcal{P})\ [det\ (2\mathcal{Q} - \mathcal{R} - \mathcal{P})]^2\ det\ (2\mathcal{R} - \mathcal{Q} - \mathcal{P}) \times$$

$$det\ \begin{pmatrix} 2\mathcal{R} - \mathcal{Q} - \mathcal{P} & 0 & 0 \\ 4\mathcal{R} & 2\mathcal{Q} + \mathcal{R} & 2\mathcal{R} + \mathcal{P} \\ 4\mathcal{Q} & 2\mathcal{R} + \mathcal{P} & 4\mathcal{Q} - \mathcal{P} \end{pmatrix}_{3 \times 3}$$

$$= [det\ (\mathcal{P} - \mathcal{Q})]^2\ det\ (3\mathcal{R} - 2\mathcal{Q} - \mathcal{P})\ [det\ (2\mathcal{Q} - \mathcal{P} - \mathcal{R})]^2\ [det\ (2\mathcal{R} - \mathcal{P} - \mathcal{Q})]^2 \times$$
$$det\ [(2\mathcal{Q} + \mathcal{R})(4\mathcal{Q} - \mathcal{P}) - (2\mathcal{R} + \mathcal{P})^2]. \quad \square$$

*3.1. Complexity of the Super Subdivision Graph $SSD_{(2,t)}(\Pi_3)$*

**Theorem 9.** *For any positive integer $t \geq 1$, the number of spanning trees of the super subdivision graph $SSD_{(2,t)}[\Pi_3]$ of the prism $\Pi_3$ is given by:* $\quad \tau[SSD_{(2,t)}(\Pi_3)] = 75\ t^5\ 2^{9t-5}$.

**Proof.** Let $\Pi_3$ be the prism with vertex set $\{u_k, v_k,\ 1 \leq k \leq 3\}$. The super subdivision graph $SSD_{(2,t)}[\Pi_3]$ of the prism $\Pi_3$ has a vertex set $V[SSD_{(2,t)}(\Pi_3)] = \{u_k, v_k, u_k^j, v_k^j,\ 1 \leq k \leq 3, 1 \leq j \leq t\}$. Thus, the graph $V[SSD_{(2,t)}(\Pi_3)]$ has $\alpha = 3(3t + 2)$ vertices, see Figure 4.

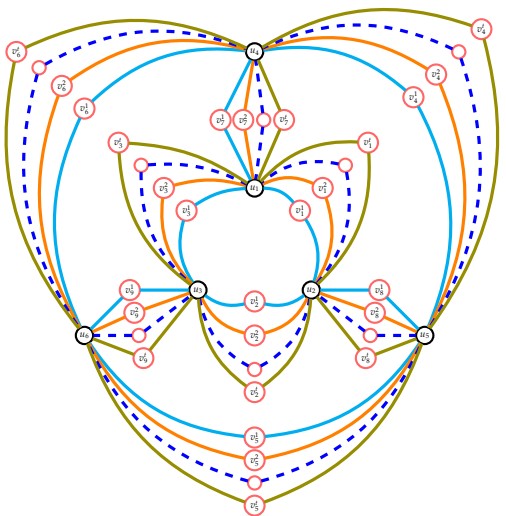

**Figure 4.** Super subdivision $SSD_{(2,t)}(\Pi_3)$ of the prism $\Pi_3$.

Applying Lemma 1, we have:

$$\tau[SSD_{(2,t)}(\Pi_3)] = \frac{1}{(9t+6)^2} \, det \, [\, (9t+6)I - \overline{D} + \overline{A}\,] = \frac{1}{(9t+6)^2} \, det \begin{pmatrix} \Phi_{(6\times6)} & \Delta_{(6\times9t)} \\ \Delta^t & \Psi_{(9t\times9t)} \end{pmatrix}$$

$$= \frac{1}{(9t+6)^2}[\, det(\Phi) \times det\,(\Psi - \Delta^t\Phi^{-1}\Delta)\,], \qquad (22)$$

where $\Psi = \begin{pmatrix} \Theta & J & J & \cdots & \cdots & J \\ J & \Theta & J & \cdots & \cdots & J \\ \vdots & \cdots & \cdots & \ddots & \ddots & \vdots \\ J & J & \cdots & \cdots & \Theta & J \\ J & J & \cdots & \cdots & J & \Theta \end{pmatrix}$, $\Theta = \begin{pmatrix} 3 & 1 & 1 & \cdots & \cdots & 1 \\ 1 & 3 & 1 & \cdots & \cdots & 1 \\ \vdots & \cdots & \cdots & \ddots & \ddots & \vdots \\ 1 & 1 & \cdots & \cdots & 3 & 1 \\ 1 & 1 & \cdots & \cdots & 1 & 3 \end{pmatrix}_{t\times t}$, and $J$ is a $t \times t$

unit matrix.

$$det\,(\Phi) = det \begin{pmatrix} 3t+1 & 1 & 1 & \cdots & \cdots & 1 \\ 1 & 3t+1 & 1 & \cdots & \cdots & 1 \\ \vdots & & \cdots & \ddots & \ddots & \vdots \\ 1 & \cdots & \cdots & 1 & 3t+1 & 1 \\ 1 & 1 & \cdots & \cdots & 1 & 3t+1 \end{pmatrix}_{6\times6} = (3t+6)\,(3t)^5. \quad (23)$$

Applying Lemma 6, we have

$$det\,(\Psi - \Delta^t\Phi^{-1}\Delta) = det \begin{pmatrix} \mathcal{P} & \mathcal{Q} & \mathcal{Q} & \mathcal{R} & \mathcal{R} & \mathcal{R} & \mathcal{Q} & \mathcal{Q} & \mathcal{R} \\ \mathcal{Q} & \mathcal{P} & \mathcal{Q} & \mathcal{R} & \mathcal{R} & \mathcal{R} & \mathcal{R} & \mathcal{Q} & \mathcal{Q} \\ \mathcal{Q} & \mathcal{Q} & \mathcal{P} & \mathcal{R} & \mathcal{R} & \mathcal{R} & \mathcal{Q} & \mathcal{R} & \mathcal{Q} \\ \mathcal{R} & \mathcal{R} & \mathcal{R} & \mathcal{P} & \mathcal{Q} & \mathcal{Q} & \mathcal{Q} & \mathcal{Q} & \mathcal{R} \\ \mathcal{R} & \mathcal{R} & \mathcal{R} & \mathcal{Q} & \mathcal{P} & \mathcal{Q} & \mathcal{R} & \mathcal{Q} & \mathcal{Q} \\ \mathcal{R} & \mathcal{R} & \mathcal{R} & \mathcal{Q} & \mathcal{Q} & \mathcal{P} & \mathcal{Q} & \mathcal{R} & \mathcal{Q} \\ \mathcal{Q} & \mathcal{R} & \mathcal{Q} & \mathcal{Q} & \mathcal{R} & \mathcal{Q} & \mathcal{P} & \mathcal{R} & \mathcal{R} \\ \mathcal{Q} & \mathcal{Q} & \mathcal{R} & \mathcal{Q} & \mathcal{Q} & \mathcal{R} & \mathcal{R} & \mathcal{P} & \mathcal{R} \\ \mathcal{R} & \mathcal{Q} & \mathcal{Q} & \mathcal{R} & \mathcal{Q} & \mathcal{Q} & \mathcal{R} & \mathcal{R} & \mathcal{P} \end{pmatrix}$$

$$= [det\,(\mathcal{P} - \mathcal{Q})]^2 \, det\,(3\mathcal{R} - 2\mathcal{Q} - \mathcal{P})\,[det\,(2\mathcal{Q} - \mathcal{P} - \mathcal{R})]^2 \times$$
$$[det\,(2\mathcal{R} - \mathcal{P} - \mathcal{Q})]^2 \, det\,[(2\mathcal{Q} + \mathcal{R})(4\mathcal{Q} - \mathcal{P}) - (2\mathcal{R} + \mathcal{P})^2],$$

where $\mathcal{P} = \frac{1}{(9t^2+18t)^t} \begin{pmatrix} 27t^2+42t-8 & 9t^2+6t-8 & 9t^2+6t-8 & \cdots & \cdots & 9t^2+6t-8 \\ 9t^2+6t-8 & 27t^2+42t-8 & 9t^2+6t-8 & \cdots & \cdots & 9t^2+6t-8 \\ \vdots & \cdots & \cdots & \ddots & \ddots & \vdots \\ 9t^2+6t-8 & 9t^2+6t-8 & \cdots & \cdots & 27t^2+42t-8 & 9t^2+6t-8 \\ 9t^2+6t-8 & 9t^2+6t-8 & \cdots & \cdots & 9t^2+6t-8 & 27t^2+42t-8 \end{pmatrix}_{t\times t}$,

$\mathcal{Q} = [a_{ij}]_{t\times t}$, in which $a_{ij} = \dfrac{9t^2+9t-2}{9t^2+18t}$, and $\mathcal{R} = [b_{ij}]_{t\times t}$, in which $b_{ij} = \dfrac{9t^2+12t+4}{9t^2+18t}$.

$$det\,(\mathcal{P} - \mathcal{Q}) = (\frac{-1}{3t})^t \, det \begin{pmatrix} 1-6t & 1 & 1 & \cdots & \cdots & 1 \\ 1 & 1-6t & 1 & \cdots & \cdots & 1 \\ \vdots & \vdots & \cdots & \cdots & \ddots & \vdots \\ 1 & 1 & \cdots & \cdots & 1-6t & 1 \\ 1 & 1 & \cdots & \cdots & 1 & 1-6t \end{pmatrix}_{t\times t} = \frac{5\,2^{t-1}}{3}. \quad (24)$$

$$det\,(2\mathcal{R} - \mathcal{Q} - \mathcal{P}) = (\frac{1}{t})^t\,det\begin{pmatrix} -2t+1 & 1 & \dots & \dots & 1 & 1 \\ 1 & -2t+1 & \dots & \dots & 1 & 1 \\ \vdots & \vdots & \dots & \ddots & \vdots & \vdots \\ 1 & 1 & \dots & \dots & -2t+1 & 1 \\ 1 & 1 & \dots & \dots & 1 & -2t+1 \end{pmatrix}_{t\times t} = (-1)^t\,2^{t-1}. \quad (25)$$

$$det\,(3\mathcal{R} - 2\mathcal{Q} - \mathcal{P}) = (\frac{1}{3t})^t\,det\begin{pmatrix} -6t+4 & 4 & \dots & \dots & 4 & 4 \\ 4 & -6t+4 & \dots & \dots & 4 & 4 \\ \vdots & \vdots & \dots & \ddots & \vdots & \vdots \\ 4 & 4 & \dots & \dots & -6t+4 & 4 \\ 4 & 4 & \dots & \dots & 4 & -6t+4 \end{pmatrix}_{t\times t}$$

$$= \frac{-2t}{(3t)^t(-6t)}det\begin{pmatrix} -6t & 0 & \dots & \dots & 0 \\ 0 & -6t & 0 & \dots & 0 \\ \vdots & \vdots & \dots & \ddots & \vdots \\ 0 & \dots & \dots & -6t & 0 \\ 0 & \dots & \dots & 0 & -6t \end{pmatrix} = \frac{(-2)^t}{3}. \quad (26)$$

$$det\,(2\mathcal{Q} - \mathcal{R} - \mathcal{P}) = det\begin{pmatrix} -2 & 0 & 0 & \dots & \dots & 0 \\ 0 & -2 & 0 & \dots & \dots & 0 \\ \vdots & \vdots & \dots & \dots & \ddots & \vdots \\ 0 & 0 & \dots & \dots & -2 & 0 \\ 0 & 0 & \dots & \dots & 0 & -2 \end{pmatrix}_{t\times t} = (-2)^t. \quad (27)$$

$$det\,[(2\mathcal{Q} + \mathcal{R})(4\mathcal{Q} - \mathcal{P}) - (2\mathcal{R} + \mathcal{P})^2] = \frac{(-18)^t\,(t+2)^t}{(3t+6)^{2t}} \times$$

$$det\begin{pmatrix} 11t+14 & 9t+10 & \dots & \dots & 9t+10 & 9t+10 \\ 9t+10 & 11t+14 & \dots & \dots & 9t+10 & 9t+10 \\ \vdots & \vdots & \dots & \ddots & \vdots & \vdots \\ 9t+10 & 9t+10 & \dots & \dots & 11t+14 & 9t+10 \\ 9t+10 & 9t+10 & \dots & \dots & 9t+10 & 11t+14 \end{pmatrix}_{t\times t}$$

$$= \frac{(-18)^t\,(t+2)^t\,(9t^2+12t+4)}{(3t+6)^{2t}(2t+4)}\,det\begin{pmatrix} 2t+4 & 0 & \dots & \dots & 0 \\ 0 & 2t+4 & 0 & \dots & 0 \\ \vdots & \vdots & \dots & \ddots & \vdots \\ 0 & \dots & \dots & 2t+4 & 0 \\ 0 & \dots & \dots & 0 & 2t+4 \end{pmatrix}$$

$$= \frac{(-1)^t(2)^{2t-1}(3t+2)^2}{t+2}. \quad (28)$$

From substituting Equations (23)–(28) into Equation (22), we obtain the result. □

*3.2. Complexity of the Super Subdivision Graph $SSD_{(2,t)}(\Pi_4)$*

**Lemma 7.** *Suppose $\mathcal{P}$, $\mathcal{Q}$ and $\mathcal{R}$ are $t \times t$ block matrices and*

$$E = \begin{pmatrix} \mathcal{P} & \mathcal{Q} & \mathcal{R} & \mathcal{Q} & \mathcal{R} & \mathcal{R} & \mathcal{R} & \mathcal{R} & \mathcal{Q} & \mathcal{Q} & \mathcal{R} & \mathcal{R} \\ \mathcal{Q} & \mathcal{P} & \mathcal{Q} & \mathcal{R} & \mathcal{R} & \mathcal{R} & \mathcal{R} & \mathcal{R} & \mathcal{R} & \mathcal{Q} & \mathcal{Q} & \mathcal{R} \\ \mathcal{R} & \mathcal{Q} & \mathcal{P} & \mathcal{Q} & \mathcal{R} & \mathcal{R} & \mathcal{R} & \mathcal{R} & \mathcal{R} & \mathcal{R} & \mathcal{Q} & \mathcal{Q} \\ \mathcal{Q} & \mathcal{R} & \mathcal{Q} & \mathcal{P} & \mathcal{R} & \mathcal{R} & \mathcal{R} & \mathcal{R} & \mathcal{Q} & \mathcal{R} & \mathcal{R} & \mathcal{Q} \\ \mathcal{R} & \mathcal{R} & \mathcal{R} & \mathcal{R} & \mathcal{P} & \mathcal{Q} & \mathcal{R} & \mathcal{Q} & \mathcal{Q} & \mathcal{Q} & \mathcal{R} & \mathcal{R} \\ \mathcal{R} & \mathcal{R} & \mathcal{R} & \mathcal{R} & \mathcal{Q} & \mathcal{P} & \mathcal{Q} & \mathcal{R} & \mathcal{R} & \mathcal{Q} & \mathcal{Q} & \mathcal{R} \\ \mathcal{R} & \mathcal{R} & \mathcal{R} & \mathcal{R} & \mathcal{R} & \mathcal{Q} & \mathcal{P} & \mathcal{Q} & \mathcal{R} & \mathcal{R} & \mathcal{Q} & \mathcal{Q} \\ \mathcal{R} & \mathcal{R} & \mathcal{R} & \mathcal{R} & \mathcal{Q} & \mathcal{R} & \mathcal{Q} & \mathcal{P} & \mathcal{R} & \mathcal{R} & \mathcal{R} & \mathcal{Q} \\ \mathcal{Q} & \mathcal{R} & \mathcal{R} & \mathcal{Q} & \mathcal{Q} & \mathcal{R} & \mathcal{R} & \mathcal{R} & \mathcal{P} & \mathcal{R} & \mathcal{R} & \mathcal{R} \\ \mathcal{Q} & \mathcal{Q} & \mathcal{R} & \mathcal{R} & \mathcal{Q} & \mathcal{Q} & \mathcal{R} & \mathcal{R} & \mathcal{R} & \mathcal{P} & \mathcal{R} & \mathcal{R} \\ \mathcal{R} & \mathcal{Q} & \mathcal{Q} & \mathcal{R} & \mathcal{R} & \mathcal{Q} & \mathcal{Q} & \mathcal{R} & \mathcal{R} & \mathcal{R} & \mathcal{P} & \mathcal{R} \\ \mathcal{R} & \mathcal{R} & \mathcal{Q} & \mathcal{Q} & \mathcal{R} & \mathcal{R} & \mathcal{Q} & \mathcal{Q} & \mathcal{R} & \mathcal{R} & \mathcal{R} & \mathcal{P} \end{pmatrix}_{12\times 12}.$$

Then, $\det E = [\det (\mathcal{P} - \mathcal{R})]^3 [\det (3\mathcal{R} - 2\mathcal{Q} - \mathcal{P})]^3 [\det (2\mathcal{Q} - \mathcal{P} - \mathcal{R})]^5 \times \det (7\mathcal{R} + \mathcal{P} + 4\mathcal{Q}).$

**Proof.** Using the properties of determinants and matrix row and column operations yields:

$$\det E = \det(\mathcal{P} - \mathcal{R})\det \begin{vmatrix} \mathcal{R}-\mathcal{P} & 2\mathcal{R}-2\mathcal{Q} & 0 & 0 & 0 & 0 & 0 & 0 & 0 & 0 & 0 \\ \mathcal{R}-\mathcal{Q} & \mathcal{R}-\mathcal{P} & \mathcal{R}-\mathcal{Q} & 0 & 0 & 0 & 0 & 0 & 0 & 0 & 0 \\ 0 & 2\mathcal{R}-2\mathcal{Q} & \mathcal{R}-\mathcal{P} & 0 & 0 & 0 & 0 & 0 & 0 & 0 & 0 \\ \mathcal{R} & \mathcal{R} & \mathcal{R} & \mathcal{R}+\mathcal{P} & \mathcal{R}+\mathcal{Q} & 2\mathcal{R} & \mathcal{R}+\mathcal{Q} & \mathcal{Q} & \mathcal{Q} & \mathcal{R} & \mathcal{R} \\ \mathcal{R} & \mathcal{R} & \mathcal{R} & \mathcal{R}+\mathcal{Q} & \mathcal{R}+\mathcal{P} & \mathcal{R}+\mathcal{Q} & 2\mathcal{R} & \mathcal{R} & \mathcal{Q} & \mathcal{Q} & \mathcal{R} \\ \mathcal{R} & \mathcal{R} & \mathcal{R} & 2\mathcal{R} & \mathcal{R}+\mathcal{Q} & \mathcal{R}+\mathcal{P} & \mathcal{R}+\mathcal{Q} & \mathcal{R} & \mathcal{R} & \mathcal{Q} & \mathcal{Q} \\ \mathcal{R} & \mathcal{R} & \mathcal{R} & \mathcal{R}+\mathcal{Q} & 2\mathcal{R} & \mathcal{R}+\mathcal{Q} & \mathcal{R}+\mathcal{P} & \mathcal{Q} & \mathcal{R} & \mathcal{R} & \mathcal{Q} \\ \mathcal{R} & \mathcal{R} & \mathcal{Q} & 2\mathcal{Q} & 2\mathcal{R} & 2\mathcal{R} & 2\mathcal{Q} & \mathcal{P} & \mathcal{R} & \mathcal{R} & \mathcal{R} \\ \mathcal{Q} & \mathcal{R} & \mathcal{R} & 2\mathcal{Q} & 2\mathcal{Q} & 2\mathcal{R} & 2\mathcal{R} & \mathcal{R} & \mathcal{P} & \mathcal{R} & \mathcal{R} \\ \mathcal{Q} & \mathcal{Q} & \mathcal{R} & 2\mathcal{R} & 2\mathcal{Q} & 2\mathcal{Q} & 2\mathcal{R} & \mathcal{R} & \mathcal{R} & \mathcal{P} & \mathcal{R} \\ \mathcal{R} & \mathcal{Q} & \mathcal{Q} & 2\mathcal{R} & 2\mathcal{R} & 2\mathcal{Q} & 2\mathcal{Q} & \mathcal{R} & \mathcal{R} & \mathcal{R} & \mathcal{P} \end{vmatrix}_{11\times 11}$$

$$= \det(\mathcal{P} - \mathcal{R})^2 \det \begin{vmatrix} \mathcal{R}-\mathcal{P} & 2\mathcal{R}-2\mathcal{Q} & 0 & 0 & 0 & 0 & 0 & 0 & 0 & 0 \\ 2\mathcal{R}-2\mathcal{Q} & \mathcal{R}-\mathcal{P} & 0 & 0 & 0 & 0 & 0 & 0 & 0 & 0 \\ \mathcal{R} & \mathcal{R} & \mathcal{R}+\mathcal{P} & \mathcal{R}+\mathcal{Q} & 2\mathcal{R} & \mathcal{R}+\mathcal{Q} & \mathcal{Q} & \mathcal{Q} & \mathcal{R} & \mathcal{R} \\ \mathcal{R} & \mathcal{R} & \mathcal{R}+\mathcal{Q} & \mathcal{R}+\mathcal{P} & \mathcal{R}+\mathcal{Q} & 2\mathcal{R} & \mathcal{R} & \mathcal{Q} & \mathcal{Q} & \mathcal{R} \\ \mathcal{R} & \mathcal{R} & 2\mathcal{R} & \mathcal{R}+\mathcal{Q} & \mathcal{R}+\mathcal{P} & \mathcal{R}+\mathcal{Q} & \mathcal{R} & \mathcal{R} & \mathcal{Q} & \mathcal{Q} \\ \mathcal{R} & \mathcal{R} & \mathcal{R}+\mathcal{Q} & 2\mathcal{R} & \mathcal{R}+\mathcal{Q} & \mathcal{R}+\mathcal{P} & \mathcal{Q} & \mathcal{R} & \mathcal{R} & \mathcal{Q} \\ \mathcal{R} & \mathcal{R}+\mathcal{Q} & 2\mathcal{Q} & 2\mathcal{R} & 2\mathcal{R} & 2\mathcal{Q} & \mathcal{P} & \mathcal{R} & \mathcal{R} & \mathcal{R} \\ \mathcal{R} & \mathcal{Q}+\mathcal{R} & 2\mathcal{Q} & 2\mathcal{Q} & 2\mathcal{R} & 2\mathcal{R} & \mathcal{R} & \mathcal{P} & \mathcal{R} & \mathcal{R} \\ \mathcal{Q} & \mathcal{Q}+\mathcal{R} & 2\mathcal{R} & 2\mathcal{Q} & 2\mathcal{Q} & 2\mathcal{R} & \mathcal{R} & \mathcal{R} & \mathcal{P} & \mathcal{R} \\ \mathcal{Q} & \mathcal{R}+\mathcal{Q} & 2\mathcal{R} & 2\mathcal{R} & 2\mathcal{Q} & 2\mathcal{Q} & \mathcal{R} & \mathcal{R} & \mathcal{R} & \mathcal{P} \end{vmatrix}_{10\times 10}$$

$$= [\det (\mathcal{P} - \mathcal{R})]^2 \det (3\mathcal{R} - 2\mathcal{Q} - \mathcal{P}) \det (\mathcal{R} - 2\mathcal{Q} + \mathcal{P}) \times$$

$$\det \begin{vmatrix} \mathcal{R}+\mathcal{P} & \mathcal{R}+\mathcal{Q} & 2\mathcal{R} & \mathcal{R}+\mathcal{Q} & \mathcal{Q} & \mathcal{Q} & \mathcal{R} & \mathcal{R} \\ \mathcal{R}+\mathcal{Q} & \mathcal{R}+\mathcal{P} & \mathcal{R}+\mathcal{Q} & 2\mathcal{R} & \mathcal{R} & \mathcal{Q} & \mathcal{Q} & \mathcal{R} \\ 2\mathcal{R} & \mathcal{R}+\mathcal{Q} & \mathcal{R}+\mathcal{P} & \mathcal{R}+\mathcal{Q} & \mathcal{R} & \mathcal{R} & \mathcal{Q} & \mathcal{Q} \\ \mathcal{R}+\mathcal{Q} & 2\mathcal{R} & \mathcal{R}+\mathcal{Q} & \mathcal{R}+\mathcal{P} & \mathcal{Q} & \mathcal{R} & \mathcal{R} & \mathcal{Q} \\ 2\mathcal{Q} & 2\mathcal{R} & 2\mathcal{R} & 2\mathcal{Q} & \mathcal{P} & \mathcal{R} & \mathcal{R} & \mathcal{R} \\ 2\mathcal{Q} & 2\mathcal{Q} & 2\mathcal{R} & 2\mathcal{R} & \mathcal{R} & \mathcal{P} & \mathcal{R} & \mathcal{R} \\ 2\mathcal{R} & 2\mathcal{Q} & 2\mathcal{Q} & 2\mathcal{R} & \mathcal{R} & \mathcal{R} & \mathcal{P} & \mathcal{R} \\ 2\mathcal{R} & 2\mathcal{R} & 2\mathcal{Q} & 2\mathcal{Q} & \mathcal{R} & \mathcal{R} & \mathcal{R} & \mathcal{P} \end{vmatrix}_{8\times 8}$$

$$= [\det (\mathcal{P} - \mathcal{R})]^2 \det (3\mathcal{R} - 2\mathcal{Q} - \mathcal{P}) \det (\mathcal{R} - 2\mathcal{Q} + \mathcal{P}) [\det (2\mathcal{Q} - \mathcal{P} - \mathcal{R})]^3 \times$$

$$\det \begin{vmatrix} \mathcal{R}+\mathcal{P} & \mathcal{R}+\mathcal{Q} & 2\mathcal{R} & \mathcal{R}+\mathcal{Q} & \mathcal{Q} \\ \mathcal{R}+3\mathcal{Q} & \mathcal{P}+\mathcal{R}+2\mathcal{Q} & 3\mathcal{R}+\mathcal{Q} & 4\mathcal{R} & 2\mathcal{R} \\ 5\mathcal{R}+\mathcal{Q} & 2\mathcal{R}+\mathcal{P}+3\mathcal{Q} & 2\mathcal{R}+\mathcal{P}+3\mathcal{Q} & 5\mathcal{R}+\mathcal{Q} & 3\mathcal{R} \\ 5\mathcal{R}+\mathcal{Q} & 5\mathcal{R}+\mathcal{Q} & 2\mathcal{R}+\mathcal{P}+3\mathcal{Q} & 2\mathcal{R}+\mathcal{P}+3\mathcal{Q} & 2\mathcal{R}+\mathcal{Q} \\ \mathcal{R}+3\mathcal{Q} & 4\mathcal{R} & 3\mathcal{R}+\mathcal{Q} & \mathcal{P}+\mathcal{R}+2\mathcal{Q} & \mathcal{P}+\mathcal{Q} \end{vmatrix}_{5\times 5}$$

$$= [\det (\mathcal{P} - \mathcal{R})]^3 [\det (3\mathcal{R} - 2\mathcal{Q} - \mathcal{P})]^2 [\det (2\mathcal{Q} - \mathcal{P} - \mathcal{R})]^4 \times$$

$$det \begin{pmatrix} \mathcal{R}+\mathcal{P} & 2\mathcal{R} & 2\mathcal{R}+2\mathcal{Q} \\ \mathcal{R}+3\mathcal{Q} & 3\mathcal{R}+\mathcal{Q} & \mathcal{P}+5\mathcal{R}+2\mathcal{Q} \\ 5\mathcal{R}+\mathcal{Q} & 2\mathcal{R}+\mathcal{P}+3\mathcal{Q} & 7\mathcal{R}+4\mathcal{Q}+\mathcal{P} \end{pmatrix}_{3\times3}$$

$$= [det\,(\mathcal{P}-\mathcal{R})]^3\,[det\,(3\mathcal{R}-2\mathcal{Q}-\mathcal{P})]^3\,[det\,(2\mathcal{Q}-\mathcal{P}-\mathcal{R})]^5\,det\,(7\mathcal{R}+\mathcal{P}+4\mathcal{Q}). \quad \square$$

**Theorem 10.** *For any positive integer $t \geq 1$, the number of spanning trees of the super subdivision graph $SSD_{(2,t)}(\Pi_4)$ of the prism $\Pi_4$ is given by:* $\quad \tau[SSD_{(2,t)}(\Pi_4)] = 3\,t^7\,2^{12t}.$

**Proof.** Applying the same methodology as in Theorem 9, we obtain:

$$\tau[SSD_{(2,t)}(\Pi_4)]\frac{1}{(12t+8)^2}[\,det(\Phi_{(8\times8)}) \times det\,(\Psi_{(12t\times12t)} - \Delta^t\Phi^{-1}\Delta_{(8\times12t)})]. \tag{29}$$

$$det\,(\Phi) = det \begin{pmatrix} 3t+1 & 1 & 1 & \dots & \dots & 1 \\ 1 & 3t+1 & 1 & \dots & \dots & 1 \\ \vdots & \dots & \dots & \ddots & \ddots & \vdots \\ 1 & 1 & \dots & \dots & 1 & 3t+1 \end{pmatrix}_{8\times8} = (3t+8)\,(3t)^7. \tag{30}$$

Applying Lemma 7, we have:

$$det\,(\Psi - \Delta^t\Phi^{-1}\Delta) = det \begin{pmatrix} \mathcal{P} & \mathcal{Q} & \mathcal{R} & \mathcal{Q} & \mathcal{R} & \mathcal{R} & \mathcal{R} & \mathcal{R} & \mathcal{Q} & \mathcal{Q} & \mathcal{R} & \mathcal{R} \\ \mathcal{Q} & \mathcal{P} & \mathcal{Q} & \mathcal{R} & \mathcal{R} & \mathcal{R} & \mathcal{R} & \mathcal{R} & \mathcal{R} & \mathcal{Q} & \mathcal{Q} & \mathcal{R} \\ \mathcal{R} & \mathcal{Q} & \mathcal{P} & \mathcal{Q} & \mathcal{R} & \mathcal{R} & \mathcal{R} & \mathcal{R} & \mathcal{R} & \mathcal{R} & \mathcal{Q} & \mathcal{Q} \\ \mathcal{Q} & \mathcal{R} & \mathcal{Q} & \mathcal{P} & \mathcal{R} & \mathcal{R} & \mathcal{R} & \mathcal{R} & \mathcal{Q} & \mathcal{R} & \mathcal{R} & \mathcal{Q} \\ \mathcal{R} & \mathcal{R} & \mathcal{R} & \mathcal{R} & \mathcal{P} & \mathcal{Q} & \mathcal{R} & \mathcal{Q} & \mathcal{Q} & \mathcal{Q} & \mathcal{R} & \mathcal{R} \\ \mathcal{R} & \mathcal{R} & \mathcal{R} & \mathcal{R} & \mathcal{Q} & \mathcal{P} & \mathcal{Q} & \mathcal{R} & \mathcal{R} & \mathcal{Q} & \mathcal{Q} & \mathcal{R} \\ \mathcal{R} & \mathcal{R} & \mathcal{R} & \mathcal{R} & \mathcal{R} & \mathcal{Q} & \mathcal{P} & \mathcal{Q} & \mathcal{R} & \mathcal{R} & \mathcal{Q} & \mathcal{Q} \\ \mathcal{R} & \mathcal{R} & \mathcal{R} & \mathcal{R} & \mathcal{Q} & \mathcal{R} & \mathcal{Q} & \mathcal{P} & \mathcal{Q} & \mathcal{R} & \mathcal{R} & \mathcal{Q} \\ \mathcal{Q} & \mathcal{R} & \mathcal{R} & \mathcal{Q} & \mathcal{Q} & \mathcal{R} & \mathcal{R} & \mathcal{Q} & \mathcal{P} & \mathcal{R} & \mathcal{R} & \mathcal{R} \\ \mathcal{Q} & \mathcal{Q} & \mathcal{R} & \mathcal{R} & \mathcal{Q} & \mathcal{Q} & \mathcal{R} & \mathcal{R} & \mathcal{R} & \mathcal{P} & \mathcal{R} & \mathcal{R} \\ \mathcal{R} & \mathcal{Q} & \mathcal{Q} & \mathcal{R} & \mathcal{R} & \mathcal{Q} & \mathcal{Q} & \mathcal{R} & \mathcal{R} & \mathcal{R} & \mathcal{P} & \mathcal{R} \\ \mathcal{R} & \mathcal{R} & \mathcal{Q} & \mathcal{Q} & \mathcal{R} & \mathcal{R} & \mathcal{Q} & \mathcal{Q} & \mathcal{R} & \mathcal{R} & \mathcal{R} & \mathcal{P} \end{pmatrix}.$$

$$= [det\,(\mathcal{P}-\mathcal{R})]^3\,[det\,(3\mathcal{R}-2\mathcal{Q}-\mathcal{P})]^3\,[det\,(2\mathcal{Q}-\mathcal{P}-\mathcal{R})]^5\,det\,(7\mathcal{R}+\mathcal{P}+4\mathcal{Q})$$

where $\mathcal{P} = \dfrac{1}{(3t^2+8t)^t} \begin{pmatrix} 9t^2+18t-4 & 3t^2+2t-4 & 3t^2+2t-4 & \dots & \dots & 3t^2+2t-4 \\ 3t^2+2t-4 & 9t^2+18t-4 & 3t^2+2t-4 & \dots & \dots & 3t^2+2t-4 \\ \vdots & \dots & \dots & \ddots & \ddots & \vdots \\ 3t^2+2t-4 & 3t^2+2t-4 & \dots & \dots & 9t^2+18t-4 & 3t^2+2t-4 \\ 3t^2+2t-4 & \dots & \dots & \dots & 3t^2+2t-4 & 9t^2+18t-4 \end{pmatrix}$,

$\mathcal{Q} = [a_{ij}]_{t\times t}$, in which $a_{ij} = \dfrac{9t^2+9t-4}{9t^2+24t}$, and $\mathcal{R} = [b_{ij}]_{t\times t}$, in which $b_{ij} = \dfrac{9t^2+12t+4}{9t^2+24t}$.

$$det\,(\mathcal{P}-\mathcal{R}) = (\frac{-2}{3t})^t det \begin{pmatrix} 1-3t & 1 & 1 & \dots & \dots & 1 \\ 1 & 1-3t & 1 & \dots & \dots & 1 \\ \vdots & \vdots & \dots & \dots & \ddots & \vdots \\ 1 & 1 & \dots & \dots & 1 & 1-3t \end{pmatrix}_{t\times t} = \frac{2^{t+1}}{3}. \tag{31}$$

$$det\,(2\mathcal{Q}-\mathcal{R}-\mathcal{P}) = det \begin{pmatrix} -2 & 0 & 0 & \dots & \dots & 0 \\ 0 & -2 & 0 & \dots & \dots & 0 \\ \vdots & \vdots & \dots & \dots & \ddots & \vdots \\ 0 & 0 & \dots & \dots & 0 & -2 \end{pmatrix}_{t\times t} = (-2)^t. \tag{32}$$

$$
\det\left(3\mathcal{R} - 2\mathcal{Q} - \mathcal{P}\right) = \left(\frac{1}{3t}\right)^t \det
\begin{pmatrix}
-6t+4 & 4 & \cdots & \cdots & 4 & 4 \\
4 & -6t+4 & \cdots & \cdots & 4 & 4 \\
\vdots & \vdots & \cdots & \ddots & \vdots & \vdots \\
4 & 4 & \cdots & \cdots & -6t+4 & 4 \\
4 & 4 & \cdots & \cdots & 4 & -6t+4
\end{pmatrix}_{t \times t}
$$

$$
= \frac{-2t}{(3t)^t(-6t)} \det
\begin{pmatrix}
-6t & 0 & \cdots & \cdots & 0 \\
0 & -6t & 0 & \cdots & 0 \\
\vdots & \vdots & \cdots & \ddots & \vdots \\
0 & \cdots & \cdots & -6t & 0 \\
0 & \cdots & \cdots & 0 & -6t
\end{pmatrix} = \frac{(-2)^t}{3}.
\tag{33}
$$

$$
\det\left(7\mathcal{R} + \mathcal{P} + 4\mathcal{Q}\right) = \frac{1}{(3t+8)^t} \det
\begin{pmatrix}
42t+58 & 36t+42 & \cdots & \cdots & 36t+42 & 36t+42 \\
36t+42 & 42t+58 & \cdots & \cdots & 36t+42 & 36t+42 \\
\vdots & \vdots & \cdots & \ddots & \vdots & \vdots \\
36t+42 & 36t+42 & \cdots & \cdots & 42t+58 & 36t+42 \\
36t+42 & 36t+42 & \cdots & \cdots & 36t+42 & 42t+58
\end{pmatrix}_{t \times t}
$$

$$
= \frac{4\,(3t+2)^2}{(3t+8)^t(6t+16)} \det
\begin{pmatrix}
6t+16 & 0 & \cdots & \cdots & 0 \\
0 & 6t+16 & 0 & \cdots & 0 \\
\vdots & \vdots & \cdots & \ddots & \vdots \\
0 & \cdots & \cdots & 6t+16 & 0 \\
0 & \cdots & \cdots & 0 & 6t+16
\end{pmatrix} = \frac{(2)^{t+1}(3t+2)^2}{3t+8}.
\tag{34}
$$

From substituting Equations (30)–(34) in Equation (29), we obtain the result. □

**Theorem 11.** *For any positive integer $t \geq 1$, then*

(i) $\tau[SSD_{(2,t)}(\Pi_3 \oplus_v K_2 \oplus_v \Pi_3)] = 9\,(5)^4\,t^{11}\,2^{19t-11}.$

(ii) $\tau[SSD_{(2,t)}(\Pi_3 \oplus_v K_2 \oplus_v \Pi_4)] = 225\,t^{13}\,2^{22t-6}.$

(iii) $\tau[SSD_{(2,t)}(\Pi_4 \oplus_v K_2 \oplus_v \Pi_4)] = 9\,t^{15}\,2^{25t-1}.$

**Proof.** Using Theorem (9), Theorem (10) and Theorem (5), we obtain:

$$
(i)\ \tau[SSD_{(2,t)}(\Pi_3 \oplus_v K_2 \oplus_v \Pi_3)] = \tau[SSD_{(2,t)}(\Pi_3)] \times \tau[SSD_{(2,t)}(K_2)] \times \tau[SSD_{(2,t)}(\Pi_3)]
$$

$$
= [\,75\,t^5\,2^{9t-5}] \times [t\,2^{t-1}] \times [75\,t^5\,2^{9t-5}] = 9\,5^4\,t^{11}\,2^{19t-11}.
$$

The proofs of (ii–iii) follow similarly, as in (i). □

## 4. Complexity of the Super Subdivision of a Cycle with a Chord

Let $P_k$ be a path with $k$ edges and $k+1$ vertices. A cycle $C_n$ with a $P_k$-chord, denoted by $C_n \divideontimes P_k$, is defined as a cycle with a path $P_k$ joining two nonconsecutive vertices of the cycle. Taking the path $P_{\frac{n}{2}} = \{v_0, v_1, \cdots, v_{\frac{n}{2}}\}$ and a cycle $C_n = \{u_1, u_2, \cdots, u_n\}$ and matching the vertices $v_0$ with $u_1$ and $v_{\frac{n}{2}}$ with $u_{\frac{n}{2}+1}$, we obtain the graph $C_n \divideontimes P_{\frac{n}{2}}$, where $V[C_n \divideontimes P_{\frac{n}{2}}] = \{u_k,\ k \in [1,n]\} \bigcup \{v_k,\ k \in [1, \frac{n}{2} - 1]\}$, see Figure 5.

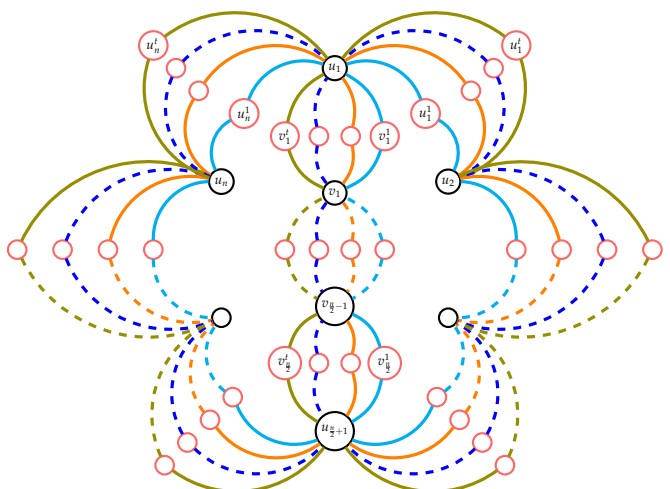

**Figure 5.** Super subdivision $SSD_{(2,t)}[C_n * P_{\frac{n}{2}}]$ of the cycle $C_n$ with a $P_{\frac{n}{2}}$ chord.

*4.1. Complexity of the Super Subdivision of $SSD_{(2,t)}(C_4 * P_2)$*

**Lemma 8.** *Suppose* $\mathcal{P}, \mathcal{Q}, \mathcal{R}$ *and* $\mathcal{S}$ *are* $t \times t$ *block matrices and* $\Lambda = \begin{pmatrix} \mathcal{P} & \mathcal{Q} & \mathcal{R} & \mathcal{S} & \mathcal{S} & \mathcal{R} \\ \mathcal{Q} & \mathcal{P} & \mathcal{S} & \mathcal{R} & \mathcal{R} & \mathcal{S} \\ \mathcal{R} & \mathcal{S} & \mathcal{P} & \mathcal{Q} & \mathcal{R} & \mathcal{S} \\ \mathcal{S} & \mathcal{R} & \mathcal{Q} & \mathcal{P} & \mathcal{S} & \mathcal{R} \\ \mathcal{S} & \mathcal{R} & \mathcal{R} & \mathcal{S} & \mathcal{P} & \mathcal{Q} \\ \mathcal{R} & \mathcal{S} & \mathcal{S} & \mathcal{R} & \mathcal{Q} & \mathcal{P} \end{pmatrix}.$

*Then,* $det\, \Lambda = det\,(\mathcal{P} + \mathcal{Q} + 2\mathcal{R} + 2\mathcal{S})\, det\,(\mathcal{P} - \mathcal{Q} - 2\mathcal{R} + 2\mathcal{S}) \times$
$det\,(\mathcal{P} - \mathcal{Q} + \mathcal{R} - \mathcal{S})^2\, det\,(\mathcal{P} + \mathcal{Q} - \mathcal{R} - \mathcal{S})^2.$

**Proof.** Using the properties of determinants and matrix row and column operations yields:

$$det\, \Lambda = det \begin{pmatrix} \mathcal{P} & \mathcal{Q} & \mathcal{R} & \mathcal{S} & 0 & 0 \\ \mathcal{Q} & \mathcal{P} & \mathcal{S} & \mathcal{R} & 0 & 0 \\ 2\mathcal{R} & 2\mathcal{S} & \mathcal{S} + \mathcal{P} & \mathcal{R} + \mathcal{Q} & 0 & 0 \\ 2\mathcal{S} & 2\mathcal{R} & \mathcal{R} + \mathcal{Q} & \mathcal{S} + \mathcal{P} & 0 & 0 \\ \mathcal{R} - \mathcal{S} & \mathcal{S} - \mathcal{R} & \mathcal{S} - \mathcal{R} & \mathcal{R} - \mathcal{S} & \mathcal{R} - \mathcal{Q} - \mathcal{S} + \mathcal{P} & 0 \\ \mathcal{R} & \mathcal{S} & \mathcal{S} & \mathcal{R} & \mathcal{R} - \mathcal{Q} & \mathcal{R} - \mathcal{Q} + \mathcal{S} - \mathcal{P} \end{pmatrix}_{6 \times 6}$$

$= det(\mathcal{R} - \mathcal{Q} + \mathcal{S} - \mathcal{P})\, det\,(\mathcal{R} - \mathcal{Q} - \mathcal{S} + \mathcal{P}) \times$

$det \begin{pmatrix} \mathcal{P} & \mathcal{P} - \mathcal{Q} & \mathcal{R} & \mathcal{R} - \mathcal{S} \\ \mathcal{P} + \mathcal{Q} & 0 & \mathcal{R} + \mathcal{S} & 0 \\ 2\mathcal{R} & 2\mathcal{R} - 2\mathcal{S} & \mathcal{S} + \mathcal{P} & \mathcal{S} + \mathcal{P} - \mathcal{R} - \mathcal{Q} \\ 2\mathcal{R} + 2\mathcal{S} & 0 & \mathcal{S} + \mathcal{P} + \mathcal{R} + \mathcal{Q} & 0 \end{pmatrix}_{4 \times 4}$

$= -det(\mathcal{R} - \mathcal{Q} + \mathcal{S} - \mathcal{P})\, det\,(\mathcal{R} - \mathcal{Q} - \mathcal{S} + \mathcal{P})\, det\,(\mathcal{P} + \mathcal{Q} + 2\mathcal{R} + 2\mathcal{S})\, det \times$

$\begin{pmatrix} \mathcal{P} - \mathcal{Q} & \mathcal{P} - \mathcal{R} & \mathcal{R} - \mathcal{S} \\ 0 & \mathcal{P} + \mathcal{Q} - \mathcal{R} - \mathcal{S} & 0 \\ 2\mathcal{R} - 2\mathcal{S} & 2\mathcal{R} - \mathcal{S} - \mathcal{P} & \mathcal{S} + \mathcal{P} - \mathcal{R} - \mathcal{Q} \end{pmatrix}_{3 \times 3}$

$= det(\mathcal{P} + \mathcal{Q} - \mathcal{R} - \mathcal{S})^2\, det\,(\mathcal{R} - \mathcal{Q} - \mathcal{S} + \mathcal{P})\, det\,(\mathcal{P} + \mathcal{Q} + 2\mathcal{R} + 2\mathcal{S}) \times$

$det \begin{pmatrix} \mathcal{P} - \mathcal{Q} & \mathcal{R} - \mathcal{S} \\ 2\mathcal{R} - 2\mathcal{S} & \mathcal{S} + \mathcal{P} - \mathcal{R} - \mathcal{Q} \end{pmatrix}_{2 \times 2}$

$= det(\mathcal{P} + \mathcal{Q} + 2\mathcal{R} + 2\mathcal{S})\, det(\mathcal{P} - \mathcal{Q} - 2\mathcal{R} + 2\mathcal{S})\, det(\mathcal{P} - \mathcal{Q} + \mathcal{R} - \mathcal{S})^2 \times$
$det(\mathcal{P} + \mathcal{Q} - \mathcal{R} - \mathcal{S})^2. \quad \square$

**Theorem 12.** *For any positive integer t, the number of spanning trees of the super subdivision graph* $SSD_{(2,t)}[C_4 \divideontimes P_2]$ *of the cycle* $C_4$ *with a* $P_2$ *chord is given by :* $\tau[SSD_{(2,t)}(C_4 \divideontimes P_2)] = 3\,t^4\,2^{6t-2}$.

**Proof.** The super subdivision graph $SSD_{(2,t)}[C_4 \divideontimes p_2]$ of the cycle $C_4$ with a $P_2$ chord has a vertex set $V[SSD_{(2,t)}(C_4 \divideontimes P_2)] = \{u_k\,, u_k^j,\ 1 \le k \in [1,4]\ \} \bigcup \{v_1, v_k^j,\ k \in [1,2]\ \}$. The graph $SSD_{(2,t)}(C_4 \divideontimes P_2)$ has $\alpha = 6t+5$ vertices and $\beta = 12t$ edges. Applying Lemma 1, we have:

$$\tau[SSD_{(2,t)}(C_4 \divideontimes P_2)] = \frac{1}{(6t+5)^2}\ det\ \begin{pmatrix} \Phi_{(5\times 5)} & \Delta_{(5\times 6t)} \\ \Delta^t & \Psi_{(6t\times 6t)} \end{pmatrix}$$

$$= \frac{1}{(6t+5)^2}[\ det(\Phi) \times det\ (\Psi - \Delta^t\Phi^{-1}\Delta)]. \tag{35}$$

$$det\ (\Phi) = det\ \begin{pmatrix} 3t+1 & 1 & 1 & 1 & 1 \\ 1 & 3t+1 & 1 & 1 & 1 \\ 1 & 1 & 2t+1 & 1 & 1 \\ 1 & 1 & 1 & 2t+1 & 1 \\ 1 & 1 & 1 & 1 & 2t+1 \end{pmatrix} = det\ \begin{pmatrix} \mathcal{M} & \mathcal{L} \\ \mathcal{L}^t & \mathcal{N} \end{pmatrix}$$

$$= det(\mathcal{M}) \times det\ (\mathcal{N} - \mathcal{L}^t\mathcal{M}^{-1}\mathcal{L}) = 12\,t^4(6t+13). \tag{36}$$

From Lemma 8, we have,

$$det\ (\Psi - \Delta^t\Phi^{-1}\Delta) = det\ \begin{pmatrix} \mathcal{P} & \mathcal{Q} & \mathcal{R} & \mathcal{S} & \mathcal{S} & \mathcal{R} \\ \mathcal{Q} & \mathcal{P} & \mathcal{S} & \mathcal{R} & \mathcal{R} & \mathcal{S} \\ \mathcal{R} & \mathcal{S} & \mathcal{P} & \mathcal{Q} & \mathcal{R} & \mathcal{S} \\ \mathcal{S} & \mathcal{R} & \mathcal{Q} & \mathcal{P} & \mathcal{S} & \mathcal{R} \\ \mathcal{S} & \mathcal{R} & \mathcal{R} & \mathcal{S} & \mathcal{P} & \mathcal{Q} \\ \mathcal{R} & \mathcal{S} & \mathcal{S} & \mathcal{R} & \mathcal{Q} & \mathcal{P} \end{pmatrix}$$

$$= det(\mathcal{P} + \mathcal{Q} + 2\mathcal{R} + 2\mathcal{S})\,det(\mathcal{P} - \mathcal{Q} - 2\mathcal{R} + 2\mathcal{S})\,det(\mathcal{P} - \mathcal{Q} + \mathcal{R} - \mathcal{S})^2 \times$$
$$det(\mathcal{P} + \mathcal{Q} - \mathcal{R} - \mathcal{S})^2,$$

where

$$\mathcal{P} = \frac{1}{(18t^2+39t)^t}\begin{pmatrix} 54t^2+93t-20 & 18t^2+15t-20 & 18t^2+15t-20 & \ldots & \ldots & 18t^2+15t-20 \\ 18t^2+15t-20 & 54t^2+93t-20 & 18t^2+15t-20 & \ldots & \ldots & 18t^2+15t-20 \\ \vdots & \ldots & \ldots & \ddots & \ddots & \vdots \\ 18t^2+15t-20 & 18t^2+15t-20 & \ldots & \ldots & 54t^2+93t-20 & 18t^2+15t-20 \\ 18t^2+15t-20 & 18t^2+15t-20 & \ldots & \ldots & 18t^2+15t-20 & 54t^2+93t-20 \end{pmatrix}_{t\times t},$$

$\mathcal{Q} = [a_{ij}]_{t\times t}$, in which $a_{ij} = \dfrac{18t^2+21t-7}{18t^2+39t}$, $\mathcal{R} = [b_{ij}]_{t\times t}$, in which $b_{ij} = \dfrac{36t^2+60t+25}{36t^2+78t}$,

and $\mathcal{S} = [c_{ij}]_{t\times t}$, in which $c_{ij} = \dfrac{36t^2+48t-1}{36t^2+78t}$.

$$det(\mathcal{P} - \mathcal{Q} + \mathcal{R} - \mathcal{S}) = det\ \begin{pmatrix} 2 & 0 & \ldots & \ldots & 0 \\ 0 & 2 & 0 & \ldots & 0 \\ \vdots & \vdots & \ldots & \ddots & \vdots \\ 0 & \ldots & \ldots & 2 & 0 \\ 0 & \ldots & \ldots & 0 & 2 \end{pmatrix} = 2^t \tag{37}$$

$$det(\mathcal{P} + \mathcal{Q} + 2\mathcal{R} + 2\mathcal{S}) = (\frac{1}{6t^2 + 13t})^t \ \times$$

$$det \begin{pmatrix} 48t^2 + 74t - 1 & 36t^2 + 48t - 1 & \dots & \dots & 36t^2 + 48t - 1 & 36t^2 + 48t - 1 \\ 36t^2 + 48t - 1 & 48t^2 + 74t - 1 & \dots & \dots & 36t^2 + 48t - 1 & 36t^2 + 48t - 1 \\ \vdots & \vdots & \dots & \ddots & \vdots & \vdots \\ 36t^2 + 48t - 1 & 36t^2 + 48t - 1 & \dots & \dots & 48t^2 + 74t - 1 & 36t^2 + 48t - 1 \\ 36t^2 + 48t - 1 & 36t^2 + 48t - 1 & \dots & \dots & 36t^2 + 48t - 1 & 48t^2 + 74t - 1 \end{pmatrix}_{t \times t} \ =$$

$$\frac{(6t + 5)^2}{(6t^2 + 13t)^t (12t + 26)} det \begin{pmatrix} 12t^2 + 26t & 0 & \dots & \dots & 0 \\ 0 & 12t^2 + 26t & 0 & \dots & 0 \\ \vdots & \vdots & \dots & \ddots & \vdots \\ 0 & \dots & \dots & 12t^2 + 26t & 0 \\ 0 & \dots & \dots & 0 & 12t^2 + 26t \end{pmatrix}$$

$$= \frac{2^{t-1} (6t + 5)^2}{(6t + 13)}. \tag{38}$$

$$det\ (\mathcal{P} - \mathcal{Q} - 2\mathcal{R} + 2\mathcal{S}) =\ det\ (\mathcal{P} + \mathcal{Q} - \mathcal{R} - \mathcal{S})$$

$$= (\frac{-1}{t})^t\ det\ \begin{pmatrix} 1 - 2t & 1 & 1 & \dots & 1 \\ 1 & 1 - 2t & 1 & \dots & 1 \\ \vdots & \dots & \dots & \ddots & \vdots \\ 1 & \dots & \dots & 1 - 2t & 1 \\ 1 & \dots & \dots & 1 & 1 - 2t \end{pmatrix}_{t \times t} = 2^{t-1}. \tag{39}$$

From substituting Equations (36)–(39) into Equation (35), we obtain the result.   □

*4.2. Complexity of the Super Subdivision of* $SSD_{(2,t)}(C_6 \divideontimes P_3)$

**Lemma 9.** *Suppose* $\mathcal{P}, \mathcal{Q}, \mathcal{R}, \mathcal{S}, \mathcal{W}, \mathcal{U}$ *and* $\mathcal{V}$ *are* $t \times t$ *block matrices and*

$$F = \begin{pmatrix} \mathcal{P} & \mathcal{Q} & \mathcal{R} & \mathcal{R} & \mathcal{S} & \mathcal{W} & \mathcal{W} & \mathcal{S} & \mathcal{R} \\ \mathcal{Q} & \mathcal{U} & \mathcal{Q} & \mathcal{S} & \mathcal{V} & \mathcal{S} & \mathcal{S} & \mathcal{V} & \mathcal{S} \\ \mathcal{R} & \mathcal{Q} & \mathcal{P} & \mathcal{W} & \mathcal{S} & \mathcal{R} & \mathcal{R} & \mathcal{S} & \mathcal{W} \\ \mathcal{R} & \mathcal{S} & \mathcal{W} & \mathcal{P} & \mathcal{Q} & \mathcal{R} & \mathcal{R} & \mathcal{S} & \mathcal{W} \\ \mathcal{S} & \mathcal{V} & \mathcal{S} & \mathcal{Q} & \mathcal{U} & \mathcal{Q} & \mathcal{S} & \mathcal{V} & \mathcal{S} \\ \mathcal{W} & \mathcal{S} & \mathcal{R} & \mathcal{R} & \mathcal{Q} & \mathcal{P} & \mathcal{W} & \mathcal{S} & \mathcal{R} \\ \mathcal{W} & \mathcal{S} & \mathcal{R} & \mathcal{R} & \mathcal{S} & \mathcal{W} & \mathcal{P} & \mathcal{Q} & \mathcal{R} \\ \mathcal{S} & \mathcal{V} & \mathcal{S} & \mathcal{S} & \mathcal{V} & \mathcal{S} & \mathcal{Q} & \mathcal{U} & \mathcal{Q} \\ \mathcal{R} & \mathcal{S} & \mathcal{W} & \mathcal{W} & \mathcal{S} & \mathcal{R} & \mathcal{R} & \mathcal{Q} & \mathcal{P} \end{pmatrix}. Then,$$

$$det\ F = det(\mathcal{P} - \mathcal{W})^2\ det(3\mathcal{R} - 2\mathcal{W} - \mathcal{P})\ det[(\mathcal{U} - \mathcal{V})(\mathcal{P} - \mathcal{W}) - 2(\mathcal{Q} - \mathcal{S})^2] \times$$
$$det[2(\mathcal{Q} + 2\mathcal{S})^2 - (\mathcal{U} + 2\mathcal{V})(3\mathcal{R} + \mathcal{P} + 2\mathcal{W})].$$

**Proof.** Performing row and column operations on the matrix combined with determinant characteristics results in:

$$det\, F = det\, \begin{pmatrix} -\mathcal{P} & -\mathcal{Q} & -\mathcal{R} & \mathcal{R} & \mathcal{S} & \mathcal{W} & 0 & 0 & 0 \\ -\mathcal{Q} & -\mathcal{U} & -\mathcal{Q} & \mathcal{S} & \mathcal{V} & \mathcal{S} & 0 & 0 & 0 \\ -\mathcal{R} & -\mathcal{Q} & -\mathcal{P} & \mathcal{W} & \mathcal{S} & \mathcal{R} & 0 & 0 & 0 \\ -2\mathcal{R} & -2\mathcal{S} & -2\mathcal{W} & \mathcal{P}+\mathcal{W} & \mathcal{Q}+\mathcal{S} & 2\mathcal{R} & 0 & 0 & 0 \\ -2\mathcal{S} & -2\mathcal{V} & -2\mathcal{S} & \mathcal{Q}+\mathcal{S} & \mathcal{U}+\mathcal{V} & \mathcal{Q}+\mathcal{S} & 0 & 0 & 0 \\ -2\mathcal{W} & -2\mathcal{S} & -2\mathcal{R} & 2\mathcal{R} & \mathcal{Q}+\mathcal{S} & \mathcal{P}+\mathcal{W} & 0 & 0 & 0 \\ 0 & 0 & 0 & 0 & 0 & 0 & \mathcal{P}-\mathcal{W} & \mathcal{Q}-\mathcal{S} & 0 \\ 0 & 0 & 0 & 0 & 0 & 0 & \mathcal{Q}-\mathcal{S} & \mathcal{U}-\mathcal{V} & \mathcal{Q}-\mathcal{S} \\ 0 & 0 & 0 & 0 & 0 & 0 & 0 & \mathcal{Q}-\mathcal{S} & \mathcal{P}-\mathcal{W} \end{pmatrix}_{9\times 9}$$

$$= det\, \begin{pmatrix} \mathcal{M} & \mathcal{O} \\ \mathcal{O} & \mathcal{N} \end{pmatrix} = det(\mathcal{M}) \times det(\mathcal{N})$$

$$= det\, \begin{pmatrix} -\mathcal{P} & -\mathcal{Q} & -\mathcal{R} & \mathcal{R} & \mathcal{S} & \mathcal{W} \\ -\mathcal{Q} & -\mathcal{U} & -\mathcal{Q} & \mathcal{S} & \mathcal{V} & \mathcal{S} \\ -\mathcal{R} & -\mathcal{Q} & -\mathcal{P} & \mathcal{W} & \mathcal{S} & \mathcal{R} \\ -2\mathcal{R} & -2\mathcal{S} & -2\mathcal{W} & \mathcal{P}+\mathcal{W} & \mathcal{Q}+\mathcal{S} & 2\mathcal{R} \\ -2\mathcal{S} & -2\mathcal{V} & -2\mathcal{S} & \mathcal{Q}+\mathcal{S} & \mathcal{U}+\mathcal{V} & \mathcal{Q}+\mathcal{S} \\ -2\mathcal{W} & -2\mathcal{S} & -2\mathcal{R} & 2\mathcal{R} & \mathcal{Q}+\mathcal{S} & \mathcal{P}+\mathcal{W} \end{pmatrix}_{6\times 6} \times det\, \begin{pmatrix} \mathcal{P}-\mathcal{W} & \mathcal{Q}-\mathcal{S} & 0 \\ \mathcal{Q}-\mathcal{S} & \mathcal{U}-\mathcal{V} & \mathcal{Q}-\mathcal{S} \\ 0 & \mathcal{Q}-\mathcal{S} & \mathcal{P}-\mathcal{W} \end{pmatrix}_{3\times 3}$$

$$= det(3\mathcal{R}-2\mathcal{W}-\mathcal{P})\ det\, \begin{pmatrix} -\mathcal{Q} & -\mathcal{R}-\mathcal{P} & \mathcal{R}+\mathcal{W} & \mathcal{S} & \mathcal{W}-\mathcal{P} \\ -\mathcal{U} & -2\mathcal{Q} & 2\mathcal{S} & \mathcal{V} & \mathcal{S}-\mathcal{Q} \\ 0 & 0 & 0 & 0 & \mathcal{P}-\mathcal{W} \\ -2\mathcal{V} & -4\mathcal{S} & 2\mathcal{Q}+2\mathcal{S} & \mathcal{U}+\mathcal{V} & \mathcal{Q}-\mathcal{S} \\ -2\mathcal{S} & -2\mathcal{W}-2\mathcal{R} & 2\mathcal{R}+\mathcal{P}+\mathcal{W} & \mathcal{Q}+\mathcal{S} & \mathcal{P}-\mathcal{W} \end{pmatrix} \times$$

$$det\, \begin{pmatrix} \mathcal{W}-\mathcal{P} & 0 & 0 \\ \mathcal{Q}-\mathcal{S} & \mathcal{U}-\mathcal{V} & 2\mathcal{Q}-2\mathcal{S} \\ 0 & \mathcal{Q}-\mathcal{S} & \mathcal{P}-\mathcal{W} \end{pmatrix}$$

$$= det\,(3\mathcal{R}-2\mathcal{W}-\mathcal{P})\, det\,(\mathcal{P}-\mathcal{W})\, det\, \begin{pmatrix} \mathcal{S}-\mathcal{Q} & \mathcal{W}-\mathcal{P} & \mathcal{R}+\mathcal{W} & \mathcal{S} \\ \mathcal{V}-\mathcal{U} & 2\mathcal{S}-2\mathcal{Q} & 2\mathcal{S} & \mathcal{V} \\ 0 & 0 & 2\mathcal{Q}+4\mathcal{S} & \mathcal{U}+2\mathcal{V} \\ 0 & 0 & 3\mathcal{R}+\mathcal{P}+2\mathcal{W} & \mathcal{Q}+2\mathcal{S} \end{pmatrix}_{4\times 4} \times$$

$$det\,(\mathcal{W}-\mathcal{P})\, det\,[(\mathcal{U}-\mathcal{V})(\mathcal{P}-\mathcal{W})-2(\mathcal{Q}-\mathcal{S})^2]$$

$$= det\,(\mathcal{P}-\mathcal{W})^2\, det\,(3\mathcal{R}-2\mathcal{W}-\mathcal{P})\, det\,[(\mathcal{U}-\mathcal{V})(\mathcal{P}-\mathcal{W})-2(\mathcal{Q}-\mathcal{S})^2] \times$$
$$det\,[2(\mathcal{Q}+2\mathcal{S})^2-(\mathcal{U}+2\mathcal{V})(3\mathcal{R}+\mathcal{P}+2\mathcal{W})]. \quad \square$$

**Theorem 13.** *For any positive integer t, the number of spanning trees of the super subdivision graph $SSD_{(2,t)}[C_6 \divideontimes P_3]$ of the cycle $C_6$ with a $P_3$ chord is given by:*
$$\tau[SSD_{(2,t)}(C_6 \divideontimes P_3)] = 3^3\, t^7\, 2^{9t-7}.$$

**Proof.** The super subdivision graph $SSD_{(2,t)}[C_6 \divideontimes p_3]$ of the cycle $C_6$ with a $P_3$ chord has a vertex set $V[SSD_{(2,t)}(C_6 \divideontimes P_3)] = \{u_k, u_k^j, 1 \le k \in [1,6]\} \bigcup \{v_k, k \in [1,2]\} \bigcup \{v_k^j, k \in [1,3]\}$. The graph $SSD_{(2,t)}(C_6 \divideontimes P_3)$ has $\alpha = 9t+8$ vertices and $\beta = 18t$ edges.

Using the same approach as in Theorem 12, we have:

$$\tau[SSD_{(2,t)}(C_6 \divideontimes P_3)] = \frac{1}{(9t+8)^2}[\,det(\Phi_{(8\times 8)}) \times det\,(\Psi_{(9t\times 9t)}-\Delta^t\Phi^{-1}\Delta_{(8\times 9t)})\,]. \quad (40)$$

$$det\ (\Phi) = 3\ (2t)^6(3t^2 + 11t). \tag{41}$$

From Lemma 9, we have,

$$det\ (\Psi - \Delta^t\Phi^{-1}\Delta) = det \begin{pmatrix} \mathcal{P} & \mathcal{Q} & \mathcal{R} & \mathcal{R} & \mathcal{S} & \mathcal{W} & \mathcal{W} & \mathcal{S} & \mathcal{R} \\ \mathcal{Q} & \mathcal{U} & \mathcal{Q} & \mathcal{S} & \mathcal{V} & \mathcal{S} & \mathcal{S} & \mathcal{V} & \mathcal{S} \\ \mathcal{R} & \mathcal{Q} & \mathcal{P} & \mathcal{W} & \mathcal{S} & \mathcal{R} & \mathcal{R} & \mathcal{S} & \mathcal{W} \\ \mathcal{R} & \mathcal{S} & \mathcal{W} & \mathcal{P} & \mathcal{Q} & \mathcal{R} & \mathcal{R} & \mathcal{S} & \mathcal{W} \\ \mathcal{S} & \mathcal{V} & \mathcal{S} & \mathcal{Q} & \mathcal{U} & \mathcal{Q} & \mathcal{S} & \mathcal{V} & \mathcal{S} \\ \mathcal{W} & \mathcal{S} & \mathcal{R} & \mathcal{R} & \mathcal{Q} & \mathcal{P} & \mathcal{W} & \mathcal{S} & \mathcal{R} \\ \mathcal{W} & \mathcal{S} & \mathcal{R} & \mathcal{R} & \mathcal{S} & \mathcal{W} & \mathcal{P} & \mathcal{Q} & \mathcal{R} \\ \mathcal{S} & \mathcal{V} & \mathcal{S} & \mathcal{S} & \mathcal{V} & \mathcal{S} & \mathcal{Q} & \mathcal{U} & \mathcal{Q} \\ \mathcal{R} & \mathcal{S} & \mathcal{W} & \mathcal{W} & \mathcal{S} & \mathcal{R} & \mathcal{R} & \mathcal{Q} & \mathcal{P} \end{pmatrix}$$

$$= det\ (\mathcal{P} - \mathcal{W})^2\ det\ (3\mathcal{R} - 2\mathcal{W} - \mathcal{P})\ det\ [(\mathcal{U} - \mathcal{V})(\mathcal{P} - \mathcal{W}) - 2(\mathcal{Q} - \mathcal{S})^2]$$
$$\times\ det\ [2(\mathcal{Q} + 2\mathcal{S})^2 - (\mathcal{U} + 2\mathcal{V})(3\mathcal{R} + \mathcal{P} + 2\mathcal{W})]$$

where

$$\mathcal{P} = \frac{1}{(36t^2 + 132t)^t} \begin{pmatrix} 108t^2 + 294t - 85 & 36t^2 + 30t - 85 & \ldots & \ldots & 36t^2 + 30t - 85 \\ 36t^2 + 30t - 85 & 108t^2 + 294t - 85 & 3\ldots & \ldots & 36t^2 + 30t - 85 \\ \vdots & & \ddots & \ddots & \vdots \\ 36t^2 + 30t - 85 & \ldots & \ldots & 108t^2 + 294t - 85 & 36t^2 + 30t - 85 \\ 36t^2 + 30t - 85 & \ldots & \ldots & 36t^2 + 30t - 85 & 108t^2 + 294t - 85 \end{pmatrix}_{t \times t},$$

$\mathcal{Q} = [a_{ij}]_{t \times t}$, in which $a_{ij} = \dfrac{3t^2 + 4t - 3}{3t^2 + 11t}$, $\mathcal{R} = [b_{ij}]_{t \times t}$, in which $b_{ij} = \dfrac{36t^2 + 60t + 25}{36t^2 + 132t}$, $\mathcal{S} = [c_{ij}]_{t \times t}$, in which $c_{ij} = \dfrac{6t^2 + 11t + 5}{6t^2 + 22t}$, $\mathcal{W} = [d_{ij}]_{t \times t}$, where $d_{ij} = \dfrac{36t^2 + 48t - 19}{36t^2 + 132t}$, $\mathcal{V} = [e_{ij}]_{t \times t}$, where $e_{ij} = \dfrac{3t^2 + 6t + 3}{3t^2 + 11t}$, and

$$\mathcal{U} = \frac{1}{(3t^2 + 11t)^t} \begin{pmatrix} 9t^2 + 25t - 8 & 3t^2 + 3t - 8 & 3t^2 + 3t - 8 & \ldots & \ldots & 3t^2 + 3t - 8 \\ 3t^2 + 3t - 8 & 9t^2 + 25t - 8 & 3t^2 + 3t - 8 & \ldots & \ldots & 3t^2 + 3t - 8 \\ \vdots & \ldots & \ldots & \ddots & \ddots & \vdots \\ 3t^2 + 3t - 8 & 3t^2 + 3t - 8 & \ldots & \ldots & 9t^2 + 25t - 8 & 3t^2 + 3t - 8 \\ 3t^2 + 3t - 8 & 3t^2 + 3t - 8 & \ldots & \ldots & 3t^2 + 3t - 8 & 9t^2 + 25t - 8 \end{pmatrix}_{t \times t}.$$

$$det\ (\mathcal{P} - \mathcal{W}) = (\frac{-1}{2t})^t\ det \begin{pmatrix} 1 - 4t & 1 & 1 & \ldots & 1 \\ 1 & 1 - 4t & 1 & \ldots & 1 \\ \vdots & \ldots & \ldots & \ddots & \vdots \\ 1 & \ldots & \ldots & 1 - 4t & 1 \\ 1 & \ldots & \ldots & 1 & 1 - 4t \end{pmatrix}_{t \times t} = 3\,2^{t-2}. \tag{42}$$

$$det\ (3\mathcal{R} - 2\mathcal{W} - \mathcal{P}) = (\frac{1}{2t})^t\ det \begin{pmatrix} -4t + 3 & 3 & \ldots & \ldots & 3 \\ 3 & -4t + 3 & \ldots & \ldots & 3 \\ \vdots & \ldots & \ddots & \vdots & \vdots \\ 3 & \ldots & \ldots & 3 & -4t + 3 \end{pmatrix}_{t \times t} = (-1)^t\,2^{t-2}. \tag{43}$$

$$det\ [(\mathcal{U} - \mathcal{V})(\mathcal{P} - \mathcal{W}) - 2(\mathcal{Q} - \mathcal{S})^2] = \frac{1}{4(t)^t}\ det \begin{pmatrix} 4t & 0 & \ldots & \ldots & 0 \\ 0 & 4t & 0 & \ldots & 0 \\ \vdots & \vdots & \ldots & \ddots & \vdots \\ 0 & \ldots & \ldots & 0 & 4t \end{pmatrix} = 2^{2t-2}. \tag{44}$$

$$det[2(\mathcal{Q} + 2\mathcal{S})^2 - (\mathcal{U} + 2\mathcal{V})(\mathcal{P} + 3\mathcal{R} + 2\mathcal{W})] = (\frac{-1}{6t^2 + 22t})^t \times$$

$$det \begin{pmatrix} 105t^2 + 208t - 24 & 81t^2 + 120t - 24 & \ldots & \ldots & 81t^2 + 120t - 24 & 81t^2 + 120t - 24 \\ 81t^2 + 120t - 24 & 105t^2 + 208t - 24 & \ldots & \ldots & 81t^2 + 120t - 24 & 81t^2 + 120t - 24 \\ \vdots & \vdots & \ldots & \ddots & \vdots & \ddots \\ 81t^2 + 120t - 24 & 81t^2 + 120t - 24 & \ldots & \ldots & 105t^2 + 208t - 24 & 81t^2 + 120t - 24 \\ 81t^2 + 120t - 24 & 81t^2 + 120t - 24 & \ldots & \ldots & 81t^2 + 120t - 24 & 105t^2 + 208t - 24 \end{pmatrix}_{t \times t}$$

$$= (\frac{-1}{6t^2 + 22t})^t \frac{(9t+8)^2}{(24t+88)} det \begin{pmatrix} 24t^2 + 88t & 0 & \ldots & \ldots & 0 \\ 0 & 24t^2 + 88t & 0 & \ldots & 0 \\ \vdots & \vdots & \ldots & \ddots & \vdots \\ 0 & \ldots & \ldots & 0 & 24t^2 + 88t \end{pmatrix}$$

$$= \frac{(-1)^t \, 2^{2t-3} \, (9t+8)^2}{3t + 11}. \tag{45}$$

From substituting Equations (41)–(45) into Equation (40), we obtain the result.  □

## 5. Complexity of the Super Subdivision of a Complete Graph

**Lemma 10.** *Suppose* $\mathcal{P}, \mathcal{Q}$ *and* $\mathcal{R}$ *are* $t \times t$ *block matrices and* $\varrho = det \begin{pmatrix} \mathcal{P} & \mathcal{Q} & \mathcal{Q} & \mathcal{Q} & \mathcal{Q} & \mathcal{R} \\ \mathcal{Q} & \mathcal{P} & \mathcal{Q} & \mathcal{R} & \mathcal{Q} & \mathcal{Q} \\ \mathcal{Q} & \mathcal{Q} & \mathcal{P} & \mathcal{Q} & \mathcal{R} & \mathcal{Q} \\ \mathcal{Q} & \mathcal{R} & \mathcal{Q} & \mathcal{P} & \mathcal{Q} & \mathcal{Q} \\ \mathcal{Q} & \mathcal{Q} & \mathcal{R} & \mathcal{Q} & \mathcal{P} & \mathcal{Q} \\ \mathcal{R} & \mathcal{Q} & \mathcal{Q} & \mathcal{Q} & \mathcal{Q} & \mathcal{P} \end{pmatrix}$.

*Then,* $\quad det \, \varrho = det(\mathcal{P} - \mathcal{R})^3 \, det(\mathcal{P} + \mathcal{R} - 2\mathcal{Q})^2 \, det(\mathcal{P} + \mathcal{R} + 4\mathcal{Q})$.

**Proof.** Using the properties of determinants and matrix row and column operations yields:

$$det \, \varrho = det \begin{pmatrix} \mathcal{P} & \mathcal{P} - \mathcal{Q} & 0 & 0 & 0 & \mathcal{P} - \mathcal{R} \\ \mathcal{Q} & \mathcal{Q} - \mathcal{P} & \mathcal{P} - \mathcal{Q} & \mathcal{P} - \mathcal{R} & \mathcal{P} - \mathcal{Q} & 0 \\ \mathcal{Q} & 0 & \mathcal{Q} - \mathcal{P} & 0 & \mathcal{Q} - \mathcal{R} & 0 \\ \mathcal{Q} & \mathcal{Q} - \mathcal{R} & \mathcal{R} - \mathcal{Q} & \mathcal{R} - \mathcal{P} & \mathcal{R} - \mathcal{Q} & 0 \\ \mathcal{Q} & 0 & \mathcal{Q} - \mathcal{R} & 0 & \mathcal{Q} - \mathcal{P} & 0 \\ \mathcal{P} + \mathcal{R} & \mathcal{P} + \mathcal{R} - 2\mathcal{Q} & 0 & 0 & 0 & 0 \end{pmatrix}_{6 \times 6}$$

$$= -det(P - R) \begin{pmatrix} \mathcal{Q} & \mathcal{Q} - \mathcal{P} & \mathcal{P} - \mathcal{Q} & \mathcal{P} - \mathcal{R} & \mathcal{P} - \mathcal{Q} \\ \mathcal{Q} & 0 & \mathcal{Q} - \mathcal{P} & 0 & \mathcal{Q} - \mathcal{R} \\ 2\mathcal{Q} & 2\mathcal{Q} - \mathcal{P} - \mathcal{R} & \mathcal{P} + \mathcal{R} - 2\mathcal{Q} & 0 & \mathcal{P} + \mathcal{R} - 2\mathcal{Q} \\ \mathcal{Q} & 0 & \mathcal{Q} - \mathcal{R} & 0 & \mathcal{Q} - \mathcal{P} \\ \mathcal{P} + \mathcal{R} & \mathcal{P} + \mathcal{R} - 2\mathcal{Q} & 0 & 0 & 0 \end{pmatrix}_{5 \times 5}$$

$$= det(\mathcal{P} - \mathcal{R})^2 \begin{pmatrix} \mathcal{Q} & 0 & \mathcal{Q} - \mathcal{P} & \mathcal{Q} - \mathcal{R} \\ 2\mathcal{Q} & 2\mathcal{Q} - \mathcal{P} - \mathcal{R} & \mathcal{P} + \mathcal{R} - 2\mathcal{Q} & \mathcal{P} + \mathcal{R} - 2\mathcal{Q} \\ \mathcal{Q} & 0 & \mathcal{Q} - \mathcal{R} & \mathcal{Q} - \mathcal{P} \\ \mathcal{P} + \mathcal{R} & \mathcal{P} + \mathcal{R} - 2\mathcal{Q} & 0 & 0 \end{pmatrix}_{4 \times 4}$$

$$= det(\mathcal{P} - \mathcal{R})^2 \, det(\mathcal{P} + \mathcal{R} - 2\mathcal{Q}) \begin{pmatrix} \mathcal{Q} & \mathcal{Q} - \mathcal{P} & \mathcal{R} - \mathcal{P} \\ \mathcal{P} + \mathcal{R} + 2\mathcal{Q} & \mathcal{P} + \mathcal{R} - 2\mathcal{Q} & 0 \\ 2\mathcal{Q} & 2\mathcal{Q} - \mathcal{R} - \mathcal{P} & 0 \end{pmatrix}_{3 \times 3}$$

$$= det(\mathcal{P} - \mathcal{R})^2 \, det(\mathcal{P} + \mathcal{R} - 2\mathcal{Q}) \, det(\mathcal{R} - \mathcal{P}) \begin{pmatrix} \mathcal{P} + \mathcal{R} + 2\mathcal{Q} & \mathcal{P} + \mathcal{R} - 2\mathcal{Q} \\ \mathcal{P} + \mathcal{R} + 4\mathcal{Q} & 0 \end{pmatrix}_{2 \times 2}$$

$$= det(\mathcal{P} - \mathcal{R})^3 \, det(\mathcal{P} + \mathcal{R} - 2\mathcal{Q})^2 \, det(\mathcal{P} + \mathcal{R} + 4\mathcal{Q}). \quad \square$$

*Complexity of the Super Subdivision Graph $SSD_{(2,t)}[K_4]$*

**Theorem 14.** *For any positive integer $t \geq 1$, the number of spanning trees of the super subdivision graph $SSD_{(2,t)}[K_4]$ of the complete graph $K_4$ is given by : $\tau[SSD_{(2,t)}(K_4)] = t^3\, 2^{6t+1}$.*

**Proof.** Let $K_4$ be a complete graph with vertex set $\{u_k\,,\ 1 \leq k \leq 4\}$. The super subdivision graph $SSD_{(2,t)}[K_4]$ of the complete graph $K_4$ has a vertex set $V[SSD_{(2,t)}(K_4)] = \{u_k\,, v_k^j,\ 1 \leq k \leq 4\,, 1 \leq j \leq t\,\}$. Thus, the graph $V[SSD_{(2,t)}(K_4)]$ has $\alpha = 2(3t+2)$ vertices and $\beta = 12t$ edges, see Figure 6.

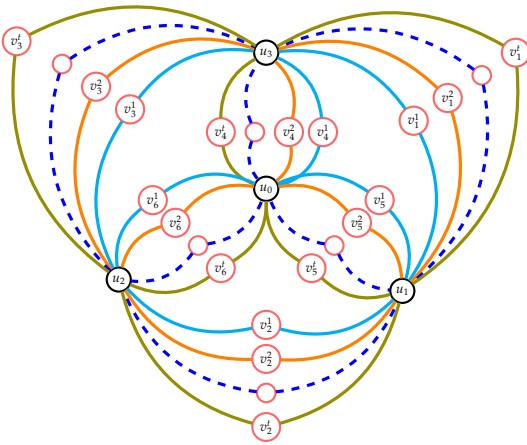

**Figure 6.** Super subdivision $SSD_{(2,t)}(K_4)$ of the complete graph $K_4$.

Applying Lemma 1, we obtain:

$$\tau[SSD_{(2,t)}(K_4)] = \frac{1}{(6t+4)^2}\, det\,[\,(6t+4)I - \overline{D} + \overline{A}\,] = \frac{1}{(6t+4)^2}\, det\,\begin{pmatrix} \Phi_{(4\times4)} & \Delta_{(4\times6t)} \\ \Delta^t & \Psi_{(6t\times6t)} \end{pmatrix}$$

$$= \frac{1}{(6t+4)^2}\,[\,det(\Phi)\,\times\,det\,(\Psi - \Delta^t\Phi^{-1}\Delta)\,]. \tag{46}$$

$$det\,(\Phi) = det\,\begin{pmatrix} 3t+1 & 1 & 1 & 1 \\ 1 & 3t+1 & 1 & 1 \\ 1 & 1 & 3t+1 & 1 \\ 1 & 1 & 1 & 3t+1 \end{pmatrix} = (3t)^3(3t+4). \tag{47}$$

From Lemma 10, we have,

$$det\,(\Psi - \Delta^t\Phi^{-1}\Delta) = det\,\begin{pmatrix} \mathcal{P} & \mathcal{Q} & \mathcal{Q} & \mathcal{Q} & \mathcal{Q} & \mathcal{R} \\ \mathcal{Q} & \mathcal{P} & \mathcal{Q} & \mathcal{R} & \mathcal{Q} & \mathcal{Q} \\ \mathcal{Q} & \mathcal{Q} & \mathcal{P} & \mathcal{Q} & \mathcal{R} & \mathcal{Q} \\ \mathcal{Q} & \mathcal{R} & \mathcal{Q} & \mathcal{P} & \mathcal{Q} & \mathcal{Q} \\ \mathcal{Q} & \mathcal{Q} & \mathcal{R} & \mathcal{Q} & \mathcal{P} & \mathcal{Q} \\ \mathcal{R} & \mathcal{Q} & \mathcal{Q} & \mathcal{Q} & \mathcal{Q} & \mathcal{P} \end{pmatrix}$$

$$= det\,(\mathcal{P} - \mathcal{R})^3\, det\,(\mathcal{P} + \mathcal{R} - 2\mathcal{Q})^2\, det\,(\mathcal{P} + \mathcal{R} + 4\mathcal{Q})\,,$$

where

$$\mathcal{P} = \frac{1}{(9t^2+12t)^t} \begin{pmatrix} 27t^2+30t-4 & 9t^2+6t-4 & 9t^2+6t-4 & \dots & \dots & 9t^2+6t-4 \\ 9t^2+6t-4 & 27t^2+30t-4 & 9t^2+6t-4 & \dots & \dots & 9t^2+6t-4 \\ \vdots & \dots & \dots & \ddots & \ddots & \vdots \\ 9t^2+6t-4 & 9t^2+6t-4 & \dots & \dots & 27t^2+30t-4 & 9t^2+6t-4 \\ 9t^2+6t-4 & 9t^2+6t-4 & \dots & \dots & 9t^2+6t-4 & 27t^2+30t-4 \end{pmatrix}_{t\times t},$$

$\mathcal{Q} = [a_{ij}]_{t\times t}$, in which $a_{ij} = \dfrac{3t+3}{3t+4}$, and $\mathcal{R} = [b_{ij}]_{t\times t}$, in which $b_{ij} = \dfrac{9t^2+12t+4}{9t^2+12t}$.

$$det\,(\mathcal{P}-\mathcal{R}) = (\frac{-2}{3t})^t det \begin{pmatrix} 1-3t & 1 & 1 & \dots & \dots & 1 \\ 1 & 1-3t & 1 & \dots & \dots & 1 \\ \vdots & \vdots & \dots & \dots & \ddots & \vdots \\ 1 & 1 & \dots & \dots & 1-3t & 1 \\ 1 & 1 & \dots & \dots & 1 & 1-3t \end{pmatrix}_{t\times t} = \frac{2^{t+1}}{3}. \quad (48)$$

$$det(\mathcal{P}+\mathcal{R}-2\mathcal{Q}) = det \begin{pmatrix} 2 & 0 & \dots & \dots & 0 \\ 0 & 2 & 0 & \dots & 0 \\ \vdots & \vdots & \dots & \ddots & \vdots \\ 0 & \dots & \dots & 0 & 2 \end{pmatrix} = 2^t. \quad (49)$$

$$det(\mathcal{P}+\mathcal{R}+4\mathcal{Q}) = (\frac{1}{3t+4})^t det \begin{pmatrix} 24t+26 & 18t+18 & \dots & \dots & 18t+18 & 18t+18 \\ 18t+18 & 24t+26 & \dots & \dots & 18t+18 & 18t+18 \\ \vdots & \vdots & \dots & \ddots & \vdots & \vdots \\ 18t+18 & 18t+18 & \dots & \dots & 24t+26 & 18t+18 \\ 18t+18 & 18t+18 & \dots & \dots & 18t+18 & 24t+26 \end{pmatrix}_{t\times t}$$

$$= \frac{2(3t+2)^2}{(3t+4)^t(6t+8)} det \begin{pmatrix} 6t+8 & 0 & \dots & \dots & 0 \\ 0 & 6t+8 & 0 & \dots & 0 \\ \vdots & \vdots & \dots & \ddots & \vdots \\ 0 & \dots & \dots & 6t+8 & 0 \\ 0 & \dots & \dots & 0 & 6t+8 \end{pmatrix} = \frac{2^t\,(3t+2)^2}{(3t+4)}. \quad (50)$$

From substituting Equations (47)–(50) into Equation (46), we obtain the result. □

## 6. Diagrammatic Comparison of the Obtained Graphs Complexities

This section includes an overview of graphical visualisations and a comparison of the values of network complexity listed in this paper. Figure 7a displays the distinct geometric shapes of the values of the complexity of the graphs generated by the super subdivision of a cycle $C_n$, where $n = 3, 4, 5, 6$. When comparing the relative complexity of these graphs, we determined that the green layer is the dominant one. Similarly, Figure 7b–d displays the distinct geometric shapes of the values of the complexity of the graphs generated by the super subdivision of a prism $\Pi_n$, the graph $(C_n \divideontimes P_{\frac{n}{2}})$ and the complete graph $K_n$, respectively.

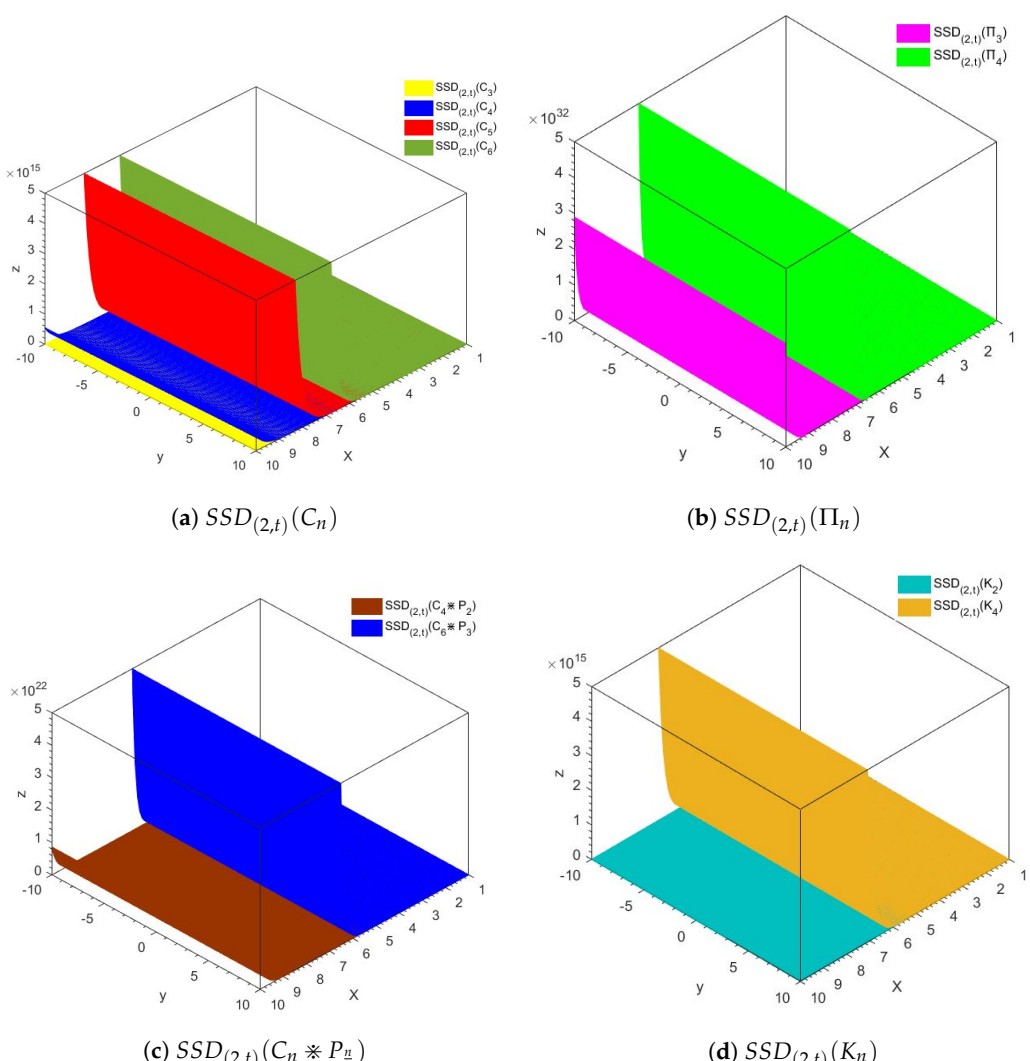

(**a**) $SSD_{(2,t)}(C_n)$      (**b**) $SSD_{(2,t)}(\Pi_n)$

(**c**) $SSD_{(2,t)}(C_n \divideontimes P_{\frac{n}{2}})$      (**d**) $SSD_{(2,t)}(K_n)$

**Figure 7.** Variations in the enumerated complexities of the super subdivision graphs.

## 7. Conclusions

One important algebraic invariant in networks (graphs) nowadays is complexity. This invariant informs us of the total number of acyclic networks in the initial network, which ultimately ensures the accuracy and dependability underlying the network. The super subdivision operation produces a more complex network. In the above work, by using the characteristics of the block matrix, we discovered straightforward and explicit formulas for determining the complexity of the super subdivision of the following graphs: the cycle $C_n$, where $n = 3, 4, 5, 6$; the dumbbell graph $Db_{m,n}$; the dragon graph $P_m(C_n)$; the prism graph $\Pi_n$, where $n = 3, 4$; a cycle $C_n$ with a $P_{\frac{n}{2}}$-chord, where $n = 4, 6$; and the complete graph $K_4$. Finally, the outcomes of our investigation were presented using 3D graphics.

**Author Contributions:** M.R.Z.E.D.: ideation, methodology, validation, formal analysis, research, writing, and review. W.A.A.: original draft writing, approach, and research. H.M.E.-S.: review and editing. The final manuscript was read by all authors, who gave their approval. All authors have read and agreed to the published version of the manuscript.

**Funding:** This research received no external funding

**Acknowledgments:** The editors' support and the anonymous reviewer's wise critique, which served to enhance the paper's presentation, are both gratefully acknowledged by the authors.

**Conflicts of Interest:** The authors declare no conflict of interest.

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
