# Peer review of "The Complexity of the Super Subdivision of Cycle-Related Graphs Using Block Matrices"

_computation, doi:10.3390/computation11080162_

Round 1

Reviewer 1 Report

Dear Professor,

Editor-in-Chief
Computation

The manuscript "The Complexity of Super Subdivision of Cycle-Related Graphs Using Block Matrices" presents an explicit formulas for determining the complexity of the super subdivision of a certain family 8 of graphs, including the cycle Cn, where n = 3, 4, 5, 6, the dumbbell graph Dbm,n, the dragon graph Pm(Cn), the prism graph Πn, where n = 3, 4, the cycle Cn with a  -chord when n = 4, 6 and the complete graph K4.

The paper has good results and suitable for publication in Computation.

Before considering further, authors must have addressed the following points:

Comments:

1-   In abstract line 3 "The best and most dependable network is the one with the most spanning trees" will be "The best and most dependable network is one of the most spanning trees".

2-   In abstract line 10, the sentence " 3D graphs will be utilized to illustrate our deductions." Is not suitable, authors must state their deductions in the abstract.

3-   The introduction must not begin with a sentence " In this paper, " and the sentence " undirected and connected graph " can be written before defining the garphs Γ = (V(Γ), E(Γ)).

4-   In introduction, line 2, the reference [1] must be stated after the definition of spanning trees.

5-   The definition of "the degree and adjacency matrices" must be presented before Lemma 1 in introduction and its citation.

6-   In Lemma 2, "the complement of Γ." Must be "where  is the complement of ".

7-   There are many typos in Definition 2 such that "Let Γ be a graph," must be "Let Γ be a graph." . "the super subdivision of Γ ," is a new sentence.

8-   In the proof of Lemma 4, authrs used "Using the properties of determinants and matrix row and column operations". These are the properties of maticies. It is also better to state properties (C1+C2, C2, C2+C3) and so on.

9-   Theorems 2, 3 and 4 are a general calculation for Theorem 1 to calculate the number of spanning trees for C4, C5 and C6. It is better to write it as corollaries without proof.

10-                 It is better to give an algorithm to connect the results in sections 2, 3,4 and 5.

11-                 What is the program that is suitable to give comparisons in Figures 8, 9 and 10.

The English Language is acceptable

Author Response

Replies to the Comments of the Reviewer(s)

The Complexity   of    Super Subdivision of Cycle-Related Graphs Using Block Matrices

The author would like to thank the referees for their comments, which led to the improvement of this paper.

There are some questions, here is the answer to these questions.

number

Before

After

1

0" from intro was supposed to be 1

has been modified

2

The list of references is short.

The connectivity of your research with other studies is low.

We update the introduction part to ensure that our research is well connected to previous studies, and we add some related and recent references to the list. 

3

Figures 7 are more or less trivial. Please consider compacting them in only one or removing them entirely. The same for Fig. 8, Fig. 9 and Fig. 10.

What is the program that is suitable to give comparisons in

Figures 8, 9 and 10.

We improved the last section  “  Diagrammatic comparison of the obtained graphs complexities “  by delete the individual figures and   merging the important  parts ( comparison figures )   of Figures 7, 8, 9, and 10 into a single, significant figure , Figures 7.   We  use  Matlab  program to  make the comparisons  of these  figures .

4

In abstract line 3 "The best and most dependable network is the

one with the most spanning trees" will be "The best and most dependable network is one of the most spanning trees". 

has been modified in line 3

5

In abstract line 10, the sentence " 3D graphs will be utilized to

illustrate our deductions." Is not suitable, authors must state

their deductions in the abstract.

has been modified

6

The introduction must not begin with a sentence " In this paper,

" and the sentence " undirected and connected graph " can be

written before defining the graphs    Γ = (V(Γ), E(Γ)).

has been modified

7

In introduction, line 2, the reference [1] must be stated after the definition of spanning trees.

has been modified in line  2

8

The definition of "the degree and adjacency matrices" must be  presented before Lemma 1 in introduction and its citation.

  We add the definitions and its citation.

9

In Lemma 2, "the complement of Γ." Must be "where  ̅ is the

complement of   ".

                  has been modified  

10

There are many typos in Definition 2 such that "Let Γ be a graph," must be "Let Γ be a graph." . "the super subdivision of  Γ ," is a new sentence.

has been modified

11

In the proof of Lemma 4, authors used "Using the properties of  determinants and matrix row and column operations". These are the properties of matrices. It is also better to state properties

(C1+C2, C2, C2+C3) and so on.

   In Lemma 4  and  Lemma 5 we added  the properties of matrices like adding C_{2} to   C_{1} and  so  on .

To clarify how we obtain the next matrix from the preceding one, the properties of matrices were added to Lemma 6.  Similar procedures can be carried out in other Lemmas.

12

Theorems 2, 3 and 4 are a general calculation for Theorem 1 to calculate the number of spanning trees for C4, C5 and C6. It is better to write it as corollaries without proof. 

We reduce the procedures in the proof of Theorems 2, 3, and 4 to avoid repetition.

13

Please provide the reason for different colors in Figure 1.

We modified all the figures (Figure 1,…,Figure 6)  to make  every color represent one of the steps of super subdivision operation.

14

"The complexity (number of spanning trees) of communication networks (graphs)" - complexity and the number of spanning trees is not the same. Nor graphs and comm. networks.

In our paper by   the complexity  we  mean  number of spanning trees  and    graphs  by networks 

Kind regards,

  • With all respect and thanks 
  • Mohamed Ramadan  Zeen El Deen
  • Associate prof. of Pure Mathematics,
  • Science faculty, Department of Mathematics,
  • Suez University. Egypt.

Reviewer 2 Report

Ref.: The Complexity of Super Subdivision of Cycle-Related Graphs 4 Using Block Matrices

I have some minor comments:

abstract - "The complexity (number of spanning trees) of communication networks (graphs)" - complexity and the number of spanning trees are not the same. Nor graphs and comm. networks.

"0" from intro was supposed to be 1.

"In this paper, we deal with a simple, undirected and connected graph Γ = (V(Γ), E(Γ))." - some arguments of why you chosen this sort of graphs. Please use Mathematics 5(4): 84 here (see Fig. 2 inside of that paper).

The list of references is short.

The connectivity of your research with other studies is low.

Please provide the reason for different colors in Figure 1.

Enumeration properties are important for special classes of graphs. Can you relate your study with Subgraphs by pair vertices?

Please argue about the interest to your study for the readers of Computation journal. You provided no references to similar studies published in the  Computation journal.

Figures 7 are more or less trivial. Please consider compacting them in only one or removing them entirely. The same for Fig. 8, Fig. 9 and Fig. 10.

Complexity of graphs is important to Molecular Topology as well. Please consider adding more related discussions to your results part of the paper.

The study is valuable and can be published after revisions.

Author Response

(The authors gave the same response as above.)

Round 2

Reviewer 2 Report

Some of the comments from previous review are still not addressed.

The connectivity of the study with other studies is still low.

Literature survey and references list are still not covering enough the topic.

Applications part must be more elaborated.

Author Response

Replies to the Comments of the Reviewer(s)

The Complexity   of    Super Subdivision of Cycle-Related Graphs Using Block Matrices

The author would like to thank the referees for their comments, which led to the improvement of this paper.

There are some questions, here is the answer to these questions.

number

Before

After

1

The connectivity of the study with other studies is still low.

 We  modified  the introduction part  and  make a connection  with the other studies 

2

The literature survey and references list are still not covering enough the topic.

We update the references part to ensure that our research is well connected to previous studies, by  adding  8 references related to our works

3

Applications part must be more elaborated

We improved the application part  to be related with  chemistry, topological complexity for selected molecular graphs, and networks   reliability .

 We highlighted the updated part in the   paper ( Pdf  format).

Thanks a lot for your cooperation and kindness.

Kind regards,

  • With all respect and thanks 
  • Mohamed Ramadan  Zeen El Deen
  • Associate prof. of Pure Mathematics,
  • Science faculty, Department of Mathematics,
  • Suez University. Egypt.
